# Breaking adsorption-energy scaling limitations of electrocatalytic nitrate reduction on intermetallic CuPd nanocubes by machine-learned insights

Qiang Gao[1,4], Hemanth Somarajan Pillai[1,4], Yang Huang[1], Shikai Liu[2], Qingmin Mu[1], Xue Han[1], Zihao Yan[1], Hua Zhou [3], Qian He [2✉], Hongliang Xin [1✉] & Huiyuan Zhu [1✉]

The electrochemical nitrate reduction reaction ($NO_3RR$) to ammonia is an essential step toward restoring the globally disrupted nitrogen cycle. In search of highly efficient electro-catalysts, tailoring catalytic sites with ligand and strain effects in random alloys is a common approach but remains limited due to the ubiquitous energy-scaling relations. With interpretable machine learning, we unravel a mechanism of breaking adsorption-energy scaling relations through the site-specific Pauli repulsion interactions of the metal $d$-states with adsorbate frontier orbitals. The non-scaling behavior can be realized on (100)-type sites of ordered B2 intermetallics, in which the orbital overlap between the hollow *N and subsurface metal atoms is significant while the bridge-bidentate *$NO_3$ is not directly affected. Among those intermetallics predicted, we synthesize monodisperse ordered B2 CuPd nanocubes that demonstrate high performance for $NO_3RR$ to ammonia with a Faradaic efficiency of 92.5% at $-0.5\ V_{RHE}$ and a yield rate of 6.25 mol h$^{-1}$ g$^{-1}$ at $-0.6\ V_{RHE}$. This study provides machine-learned design rules besides the $d$-band center metrics, paving the path toward data-driven discovery of catalytic materials beyond linear scaling limitations.

[1] Department of Chemical Engineering, Virginia Polytechnic Institute and State University, 635 Prices Fork Rd., Blacksburg, VA 24061, USA. [2] Department of Materials Science and Engineering, National University of Singapore, 9 Engineering Drive 1, 117575 Singapore, Singapore. [3] X-ray Science Division, Advanced Photon Source, Argonne National Laboratory, Lemont, IL 60439, USA. [4]These authors contributed equally: Qiang Gao, Hemanth Somarajan Pillai. ✉email: heqian@nus.edu.sg; hxin@vt.edu; huiyuanz@vt.edu

Nitrate ($NO_3^-$) is one of the most common water pollutants from a variety of sources, including agricultural runoff, industrial wastewater discharges, and animal manures[1]. To remediate $NO_3^-$ contamination, selective catalytic reduction systems have been actively pursued with the aim to harmonize the global nitrogen cycle (N-cycle)[2,3] involving the interconversion of dinitrogen ($N_2$) and reactive nitrogen species, e.g., ammonia ($NH_3$), nitrogen oxides ($NO_x$), and $NO_3^-$. In this regard, the electrocatalytic $NO_3^-$ reduction reaction ($NO_3$RR) to $NH_3$ with renewable electricity offers a practical path for restoring the disrupted N-cycle and, more importantly, a sustainable alternative to the energy-intensive Haber-Bosch process that results in 1–2% of global carbon dioxide ($CO_2$) emissions[4]. Of particular interest is the development of high-performance catalysts for handling high pH $NO_3^-$ concentrates due to less formation of toxic byproduct $NO_x$ and the growing concern of removing $NO_3^-$ from alkaline nuclear wastes[5].

Copper (Cu) has demonstrated promise for catalyzing $NO_3$RR in alkaline media with reasonable Faradaic efficiencies (FE) to $NH_3$[1], although high overpotentials are needed. Single-crystal experiments showed that $NO_3$RR on Cu is structure sensitive with (100)-oriented surface sites more active toward $NH_3$ formation than the (111) counterparts[5]. A series of random alloy electrocatalysts (e.g., CuNi[6], CuRh[7], and PtRu[8]) have been synthesized for $NO_3$RR with a general tradeoff of the partial current density and FE, arguably due to the ubiquitous adsorption-energy scaling relations. Many strategies have been visioned to circumvent such energy-scaling limitations[9] on catalytic performance, such as tuning strain[10] and ligand[11], designing bifunctional[12] or molecular single-site catalysts[13], and imposing nanoscopic confinement[14]. However, the lack of theoretical underpinning of site reactivity makes it difficult to implement those strategies. Therefore, it is imperative to develop theory-guided principles that are transformative in material design for the development of advanced catalytic systems, particularly for finding high-performance $NO_3$RR electrocatalysts toward $NH_3$. Ordered intermetallic alloys, with atomically ordered structures and well-defined compositions, have attracted extensive attention as excellent electrocatalysts for oxygen reduction[15–17], small molecules oxidation[18,19], and $CO_2$ reduction[20]. Compared with random alloys, the structural ordering of intermetallic nanocrystals endows them with unique electronic properties and chemical stability[21]. Nevertheless, the direct solution-phase synthesis of ordered intermetallic nanocrystals remains challenging and their structural effect on surface reaction kinetics is largely unexploited.

In this work, we develop a mechanistic understanding of $NO_3$RR on Cu surfaces in alkaline media with grand-canonical density functional theory (DFT) calculations. With the adsorption energies of bridge-bidentate *$NO_3$ and hollow *N as reactivity descriptors, the volcano plot very well captures the known activity trends of pure metals. The Bayesian theory of chemisorption (Bayeschem)[22], as an interpretable machine learning (ML) approach, then unravels the origin of linear scaling between *$NO_3$ and *N adsorption energies on metal surfaces and identified an intriguing mechanism to break the scaling by leveraging the site-specific Pauli repulsion of metal $d$-states with adsorbate frontier orbitals. The machine-learned insights point to the peculiar properties of (100)-oriented surface sites at ordered intermetallics of $d$ metals with a body-centered cubic (bcc) structure (B2) in which a shortened interlayer spacing along the bcc {100} direction results in significant overlap of the subsurface metal $d$-orbitals with the hollow *N $p$-orbitals while the bridge-bidentate *$NO_3$ is not directly influenced. We then synthesize ordered intermetallic B2 CuPd nanocubes terminated with (100) facets using a colloidal method. The CuPd catalyst exhibits superior $NO_3$RR performance compared with Cu and Pd

nanocubes in alkaline media, validating theoretical predictions. Specifically, these CuPd nanocubes demonstrate a $NH_3$ FE of 92.5% at −0.5 V vs. reversible hydrogen electrode (RHE) and a yield rate of 6.25 mol h$^{-1}$ g$^{-1}$ at −0.6 V vs. RHE in $NO_3$RR. Furthermore, these B2 CuPd nanocubes demonstrate high stability over 12 h electrolysis in 1 M $KNO_3$ + 1 M KOH. Bayeschem models suggest that while the upshifted $d$-band center of Cu sites at CuPd nanocubes favors the bridge-bidentate *$NO_3$ adsorption, the hollow *N is destabilized due to a dominant role of Pauli repulsion from the subsurface Pd $d$-orbitals, facilitating the protonation of N-bonded species toward $NH_3$. This study demonstrates the concept of combining interpretable ML with precision synthesis for designing catalytic systems that break adsorption-energy scaling relations and circumvent the corresponding limitations on attainable catalytic performance.

## Results

**Structure-activity relationships of $NO_3$RR on metal surfaces.** DFT calculations using the Vienna Ab initio Simulation Package (VASP) were performed to probe reaction pathways of $NO_3$RR on metal surfaces. In the $NO_3$RR literature, there are several suggested reaction pathways toward $NH_3$ formation depending on the electrolyte pH and catalysts[5,23,24], and there is no consensus regarding the critical intermediates and governing factors. Supplementary Figure 1 shows the free energy diagram of different reaction pathways on Cu(100) and Cu(111) at 0 V vs. RHE in alkaline conditions. We chose the pathway *$NO_3$ → *$NO_2$ → *NO → *NHO → *$NH_2O$ → *$NH_2OH$ → *$NH_2$ → $NH_3$ because it is the most thermodynamically favorable one on both Cu(100) and Cu(111). Figure 1a, b shows the structures of reaction intermediates and free energy profiles of $NO_3$RR to $NH_3$ on Cu(100) and Cu(111) at 0 V vs. RHE calculated from the grand-canonical DFT approach (see Supplementary Tables 1–3 for free energy corrections and source data). In alkaline media, the $NO_3$RR is assumed to follow a series of deoxidation steps to form *NO, and its further reduction to $NH_3$ which likely goes through *NHO to adsorbed hydroxylamine (*$NH_2OH$)[5]. It is generally accepted that $NH_2OH$ can only be transiently observed[25], which is then readily reduced to $NH_3$. At 0 V vs. RHE, the overpotential ($E^0 = +0.69$ V vs. RHE at pH 14) drives the removal of N-bonded species through electrochemical steps. However, the adsorption of negatively charged *$NO_3$ species (~−1.0 e for both surfaces from Bader charge analysis) is thermodynamically uphill, which is consistent with the observation that the adsorption of $NO_3^-$ ions is rate-limiting for $NO_3$RR on Cu at highly reductive potentials. The more favorable adsorption of $NO_3^-$ on Cu(100) than that on Cu(111) explains an earlier onset potential for $NO_3^-$ reduction on Cu(100)[5]. Moreover, *$NO_2$ formation from $NO_2^-$ is more endergonic on Cu(111) than that on Cu(100), resulting in a slower re-adsorption of $NO_2^-$ and its further reduction on (111)-type sites as observed[24].

To understand the activity trends of $NO_3$RR across elemental metals, we have calculated the free formation energies of reaction intermediates on the (100) and (111) facets of late transition and noble metals at the most stable configurations. Linear adsorption-energy scaling relations of reaction intermediates with *$NO_3$ and *N as descriptor species (Supplementary Fig. 2), were used to develop the activity volcano plot (Fig. 1c). The adsorption energies of *N at the four-fold hollow site and *$NO_3$ at the bridge site were used because of the simplicity and accuracy for capturing the thermodynamic stability of reaction intermediates. We used the highest free energy change of all reaction steps at 0 V vs. RHE (close to experimentally measured onset potentials) to characterize the theoretical activity of (100)- and (111)-like metal sites. Although many promising approaches have been proposed to calculate electrochemical barriers[26–28], very few benchmarks

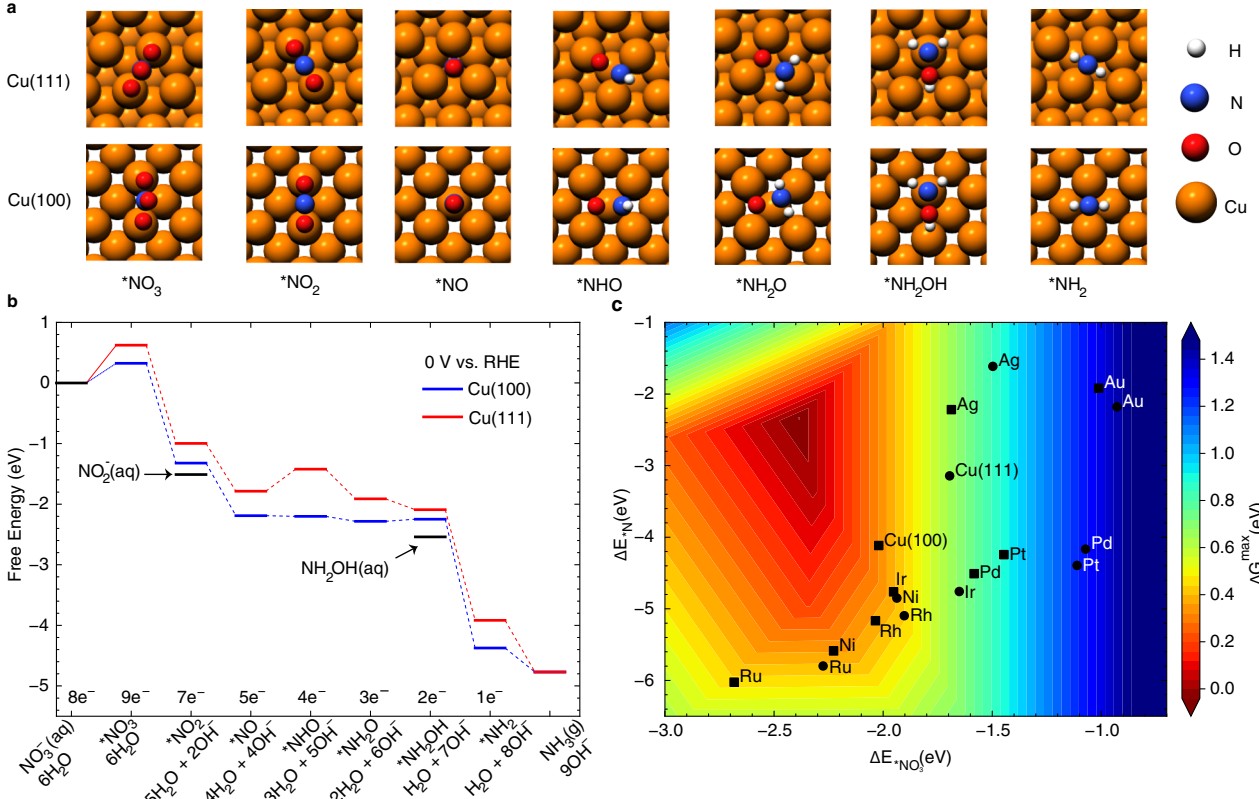

**Fig. 1 Reaction pathways of NO₃RR to NH₃ and the activity volcano plot. a** Adsorption configurations of reaction intermediates (top view) are shown (Cu: orange-red, N: blue, O: red, H: gray). **b** Free energy profiles of NH₃ formation from NO₃⁻ reduction on Cu(100) and Cu(111) at 0 V vs. RHE from grand-canonical DFT calculations. The free formation energies of intermediate products NO₂⁻(aq) and NH₂OH(aq) are marked. **c** The activity volcano plot of NO₃RR to NH₃ using adsorption energies of the bridge-bidentate *NO₃ and hollow *N as reactivity descriptors. Circle and square symbols represent (111) and (100) metal surfaces, respectively.

are available. The choice of thermodynamics-based descriptors instead of full kinetic analysis with explicitly computed activation barriers is deliberate for capturing general activity trends and guiding experimental design. The predicted activity trend of Cu(100) > Cu(111) is consistent with experimental measurements[5]. The volcano map also suggests that Cu is more active than Pd, Ag, and Au because of a favorable *NO₃ formation and facile removal of N-bonded species. It is notable that there is a strong linear scaling of adsorption energies between *NO₃ and *N for late transition metals with a slope of 1.19 (R²: 0.88), close to the theoretical slope of 1.5[29]. Coinage metals (Cu, Ag, and Au) typically follow energy scaling with a different slope due to repulsive interactions from the fully occupied metal $d$-band[30]. Nevertheless, the desire is to break such scaling relations toward the top of the activity volcano, i.e., finding the optimal catalysts that adsorb *NO₃ stronger and *N weaker than Cu(100). Although various strategies of harnessing internal and external factors in complex materials were envisioned[9,31], there is no theoretical guidance for going beyond the linear adsorption-energy scaling relations that largely limit the attainable catalytic performance.

**Physical insights from interpretable ML**. To pinpoint the physical origin of the linear scaling relations between *NO₃ and *N adsorption energies on metal surfaces beyond the valency argument, we have employed a recently-developed Bayesian theory of chemisorption (Bayeschem) as an interpretable ML approach[15]. Built upon the $d$-band theory of chemisorption and Bayesian optimization by learning from ab initio adsorption properties, Bayeschem has been used for qualitatively understanding the

nature of chemical bonding and underlying electronic factors governing the trend of surface reactivity[15]. Figure 2a shows the DFT-calculated vs. model-predicted *NO₃ and *N adsorption energies on (100)- and (111)-terminated metal surfaces. The details of model development and posterior distributions of interaction parameters can be found in Supplementary Figs. 3–6. Figure 2b shows DFT-calculated and model-predicted density of states projected onto adsorbate frontier orbitals, with Cu(100) as an example. For *N, both $p_{xy}$ and $p_z$ orbitals contribute to the adsorption energy with clearly captured bonding and antibonding states. In comparison, the HOMO of *NO₃ has antibonding states pinning across the Fermi level, while the LUMO is too high in energy to be occupied, thus forming a Lorenzian-shaped resonance state (Fig. 2b). By varying electronic factors in the Newns-Anderson model Hamiltonian[15] (using Cu as a reference), we are showing the Bayschem-predicted change of *NO₃ and *N adsorption energies in Figs. 2c, d for (100)- and (111)-like surface sites, respectively. As the $d$-band center of surface sites shifts around in response to a perturbation, for example, by +1.5 to −1.5 eV, the adsorption energies of *NO₃ and *N change side by side, i.e., a higher (lower) $d$-band center leads to a more (less) favorable interaction, irrespective of surface termination (Fig. 2c, d). Interestingly, with the increase of the interatomic coupling strength $V^2_{ad}$, *N adsorption at the (100) hollow site becomes stronger first due to the depopulation of adsorbate-metal antibonding states but weakens as the change of $V^2_{ad}$ is larger than 4.0 eV² where Pauli repulsion becomes dominant. This nonlinear correlation results from a complex interplay of the orbital hybridization and orthogonalization, both of which depend on the coupling strength $V^2_{ad}$, albeit in different slopes

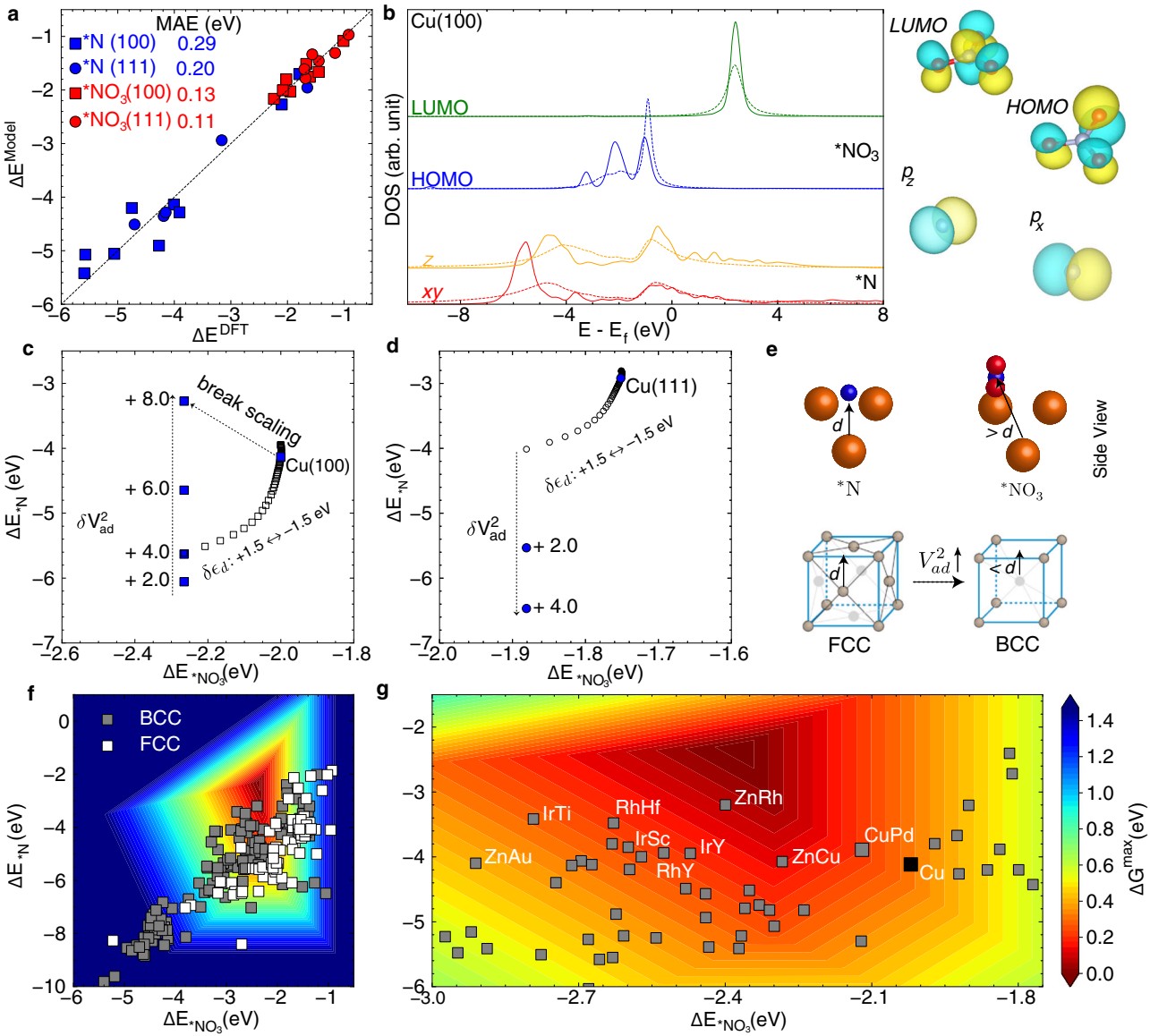

**Fig. 2 Breaking linear adsorption-energy scaling relations enabled by machine-learned physical insights. a** Bayesian models of chemisorption (Bayeschem) for $*NO_3$ and $*N$ on (100)- and (111)-terminated metal surfaces. **b** Projected density of states onto adsorbate frontier orbitals from DFT calculations (solid) and Bayeschem (dashed) model prediction, taking Cu(100) as an example. Localized wannier functions projected onto the frontier orbitals of gas-phase $NO_3$ and N radicals are shown. **c**, **d** Bayeschem-predicted adsorption energies of $*NO_3$ and $*N$ on Cu(100) and Cu(111) by perturbing the electronic structure of adsorption sites. **e** Schematic illustration of the Pauli repulsion for breaking energy-scaling relations between $*NO_3$ and $*N$ adsorption energies on (100)-oriented surfaces of B2 intermetallics due to phase-induced reduction of layer separations. **f** DFT-calculated adsorption energies of $*NO_3$ and $*N$ on (100)-terminated B2 intermetallics from the Materials Project and randomly sampled (100)-terminated fcc intermetallics. **g** DFT-calculated adsorption energies of $*NO_3$ and $*N$ on (100)-terminated B2 intermetallics close to the activity volcano top. Cu(100) and a few interesting systems (the first element denotes the surface metal) are highlighted.

(Supplementary Fig. 7a). An increase of the $V^2_{ad}$ for Cu(111) leads to a monotonically strengthened adsorption of $*N$ until its value becomes unrealistically large (Supplementary Fig. 7b). Structural analysis of 8 site motifs (Supplementary Table 4 and Fig. 8) showed that the increase of interatomic coupling can be more drastic for the B2 (bcc) intermetallic structures because the subsurface metal-ligand is close to the hollow $*N$, as illustrated in Fig. 2e. The Bayeschem model suggests that the (100)-facet of B2 structural motifs can potentially break adsorption-energy scaling relations. With the theoretical guidance from the Bayeschem model, we show the DFT-calculated adsorption energies of $*NO_3$ and $*N$ on (100)-terminated B2 intermetallics and on randomly sampled (100)-terminated face-centered cubic (fcc)

intermetallics from the Materials Project (Fig. 2f). The randomly sampled (100)-surfaces follow linear scaling relations as suggested by the Bayeschem. Compared with Cu(100), (100)-terminated B2 CuPd, ZnRh, and ZnCu structures (the first element denotes the surface metal) are predicted to exhibit reactivity properties beyond the scaling relations, i.e., a stronger $*NO_3$ adsorption and a weaker $*N$ adsorption (Fig. 2g). Zn-terminated ones are likely not stable in $NO_3RR$ operating conditions, and thus not considered for further studies. Cu-terminated CuPd (100) and (111) surfaces have been previously shown to be more stable than the Pd-termination using DFT-calculated surface energies[32,33]. DFT-calculated free energy diagrams of the full reaction pathway predict that the activity metric $\Delta G^{max}$ at CuPd(100) is ~0.10 eV

lower than that at Cu(100), with both surfaces limited by $*NO_3$ adsorption (Supplementary Fig. 9). Although DFT calculations at the GGA-PBE level have an often-quoted error of ±0.2 eV for adsorption energies at metal surfaces, the relative error across similar systems is expected to be much smaller and the qualitative prediction is reliable in the methodology. To validate the physical understanding of chemical bonding attained from machine learning, we have performed a detailed electronic structure analysis of Cu(100) and CuPd(100) in Supplementary Fig. 10. As suggested by the Bayeschem ML models, the higher $d$-center of site Cu atoms at CuPd stabilizes $*NO_3$ while the larger interatomic coupling from the subsurface Pd ligand destabilizes the $*N$, realizing an independent tuning of the $*N$ and $*NO_3$ binding energies to a certain extent. Guided by the physical insights obtained from interpretable ML instead of explicitly exploring the intermetallic design space (requiring at least 7226 DFT calculations, see Supplementary Table 4), we greatly speed up the screening process and quickly narrow down the B2 CuPd intermetallic as the target for synthesis.

**Synthesis and structural characterizations of monodisperse intermetallic CuPd nanocubes.** High-temperature annealing is usually involved in the synthesis of ordered intermetallic alloys in order to promote metal atom rearrangement and $d$-$d$ orbital hybridization[18,26–28]. Such a high-temperature process involves solid-state annealing and results in the sintering and aggregation of nanocrystals. Solution-phase synthesis of ordered intermetallic nanoparticles is more desirable and has been reported in the PdCu[34] and AuCu[21,35] systems. Unfortunately, it usually requires an additional step either electrochemically or through a seed-mediated growth and diffusion process, and the produced intermetallic alloys are partially ordered[34–36]. The direct, facile synthesis of monodisperse ordered intermetallic nanocrystals in the solution phase remains a great challenge. Herein, we have synthesized well-defined ordered intermetallic CuPd nanocubes through the consecutive decomposition and reduction of corresponding metal precursors. Briefly, 0.1 mmol [Cu(acac)$_2$] (acac = acetylacetonate), 0.1 mmol PdCl$_2$ and 0.4 mmol 1,2-tetradecanediol (TDD) were mixed in 10 mL oleylamine (OAm), and heated under N$_2$ atmosphere to 80 °C for 30 min. 0.5 mL trioctylphosphine (TOP) was injected into the solution. The mixture solution was then rapidly heated to 250 °C and maintained for 30 min. During the synthesis, TOP acts as the surfactant, while OAm acts as both the solvent and reductant. The synthesis conditions were optimized accordingly (see detailed discussion in Supplementary Information, Supplementary Figs. 12–15). Representative scanning electron microscopy (SEM) secondary-electron image (Fig. 3a) and transmission electron microscopy (TEM) bright-field (BF) image (Fig. 3b) of the as-synthesized sample reveal a uniform cubic morphology with a size of 50 ± 4 nm. The TEM BF image of intermetallic CuPd nanocubes shows a brighter contrast at the center and a darker contrast at the edge, suggesting atom enrichment at the surface or corners. The atomic ordering in the CuPd nanocube can be directly visualized using Z contrast in the atomic-resolution high-angle annular dark-field scanning transmission electron microscopy (HAADF-STEM) images (Fig. 3c, d). The corresponding Fourier-transform (FT) pattern of the CuPd nanocube in Fig. 3c inset and the atomic model overlay in Fig. 3d indicated that the ordered CuPd nanocube is a single crystalline with a B2 intermetallic cubic structure. A HAADF-STEM image (Fig. 3c) and a high-resolution TEM (HRTEM) image (Supplementary Fig. 16a) show that the particles are single-crystalline with a lattice spacing of 2.95 Å, which can be assigned to the (100) planes of the ordered CuPd intermetallic phase. The selected area

electron diffraction (SAED) pattern of the CuPd nanocubes depicted in Supplementary Fig. 16b shows bright concentric rings that can be assigned to the (100), (110), and (200) planes of the ordered CuPd intermetallic phase, respectively. Moreover, as shown in Supplementary Fig. 17, the alternating intensity profile in the corresponding HAADF line profile further confirms the Cu/Pd atomic ordering within the CuPd nanocube. To get further insight into the distribution of Pd and Cu in the as-synthesized CuPd nanocubes, elemental analysis was carried out. Figure 3e shows a HAADF-STEM image and the corresponding elemental maps of a representative CuPd nanocube. We observed that Cu and Pd are homogeneously distributed across the CuPd nanocube, which is consistent with a B2 intermetallic structure.

As shown in Fig. 3f, the powder X-ray diffraction (XRD) pattern of the ordered CuPd nanocube demonstrates that the distinct peaks are completely consistent with the ordered CuPd B2 intermetallic phase (ICDD No. 01-078-4406)[17]. The appearance of characteristic superlattice peaks at $2\theta = 30°$ and 43° confirms the structure of the B2 intermetallic phase. The composition of the ordered CuPd nanocubes was further investigated by using energy-dispersive X-ray spectroscopy (EDX) and inductively coupled plasma mass spectrometry (ICP-MS), and the obtained consistent results suggested that the Pd/Cu molar ratio is about 1:1. Only Pd and Cu could be detected in the EDX spectrum except for the carbon signal which comes from the carbon-coated TEM grid (Supplementary Fig. 18). The electronic interactions between Pd and Cu were comprehensively investigated via multiple characterization techniques. X-ray photoelectron spectroscopy (XPS) measurements show that the binding energy of Pd $3d$ core levels upshifts by ~0.68 eV versus pure Pd nanocubes (Fig. 4a), while the binding energy of Cu $2p$ core levels decreases by ~0.80 eV versus pure Cu nanocubes (Fig. 4b), indicating the existence of charge transfer between Pd and Cu[37,38]. This charge transfer is further confirmed by the shift of absorption edge to the higher energy direction in X-ray absorption near-edge spectroscopy (XANES) of the Pd K-edge (Fig. 4c and inset) and the Cu K-edge (Supplementary Fig. 19a). Figure 4d and Supplementary Fig. 20 present Fourier-transformed Pd K-edge extended X-ray absorption fine structure (EXAFS) spectra of the ordered CuPd nanocubes as well as the reference (Pd foil). As shown in Fig. 4d, in comparison with Pd foil, the ordered CuPd nanocubes exhibit shorter interatomic distance $R_{Pd-Cu(Pd)}$ than that of cubic close-packed Pd, providing a structural basis for employing the Pauli repulsion to weaken the $*N$ adsorption to promote NO$_3$RR catalysis. Supplementary Fig. 19b presents Fourier-transformed Cu K-edge EXAFS spectra of the ordered CuPd nanocubes and the reference (Cu foil). In comparison with Cu foil, the ordered CuPd nanocubes exhibit slightly longer interatomic distance $R_{Cu-Pd(Cu)}$ than that of cubic close-packed Cu. The shift of the first nearest coordination peaks of the ordered CuPd nanocubes demonstrates the slight change in interatomic distances (Fig. 4d). The fitting results of ordered CuPd nanocubes and Pd foil are listed in Supplementary Table 5, showing the smaller coordinate number of Pd atoms in ordered B2 CuPd nanocubes than that in bulk Pd foil, which is consistent with the theoretical coordination number of 8 for B2 structure and 12 for fcc structure. Moreover, the specific PdCu(Pd) bond length decreases from 2.74 Å for Pd foil to 2.62 Å for ordered CuPd nanocubes. Because of the reduced bond distance of CuPd, the B2-ordered intermetallic CuPd nanocubes provide a platform for leveraging Pauling repulsion to go beyond the energy-scaling relations discussed in our computational results.

In order to understand the formation mechanism of the ordered CuPd nanocubes, the nucleation and growth process were monitored through time-dependent experiments. The aliquots taken from the synthesis process at different time intervals were

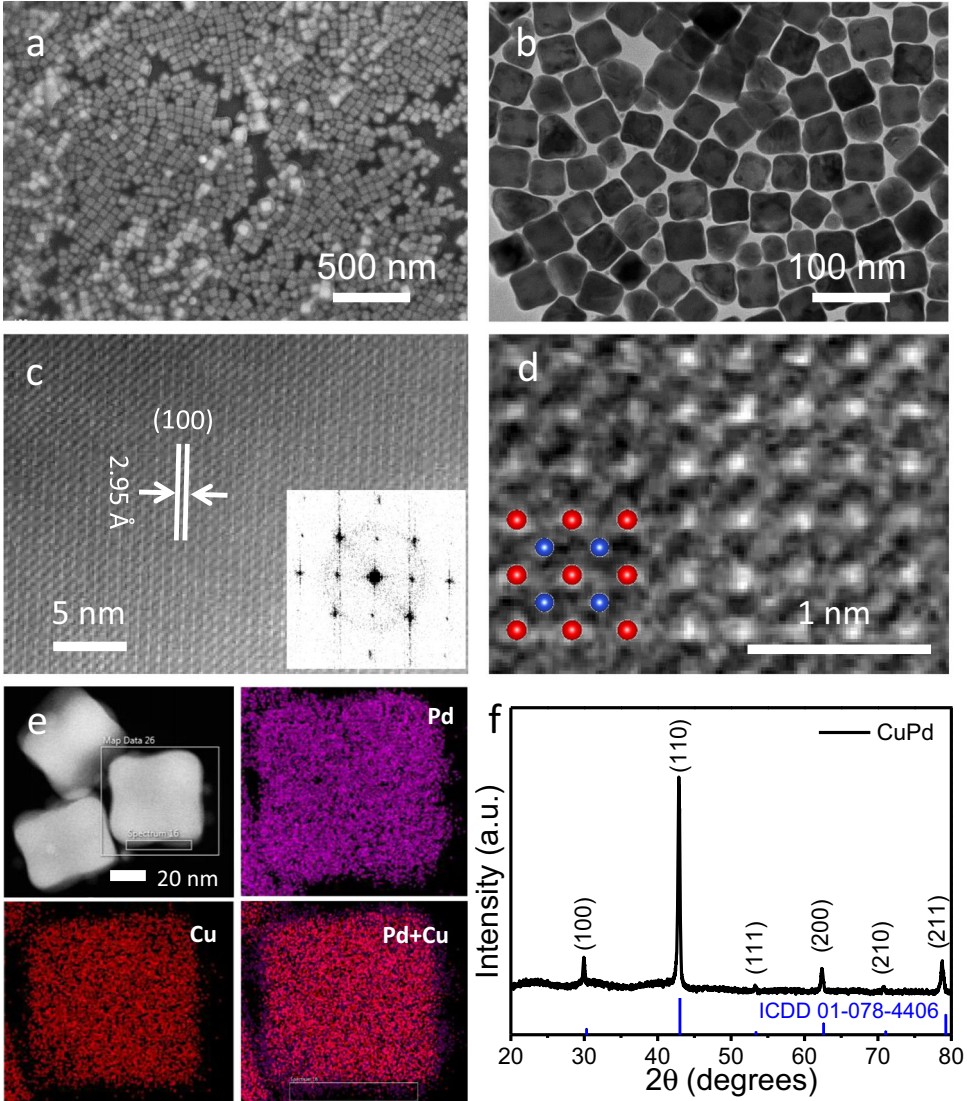

**Fig. 3 Characterizations of the B2-ordered intermetallic CuPd nanocubes. a** An SEM secondary-electron image, **b** A TEM BF image, and **c** An HAADF-STEM image of the as-synthesized ordered CuPd nanocubes. Inset shows the corresponding Fourier-transform (FT) pattern of the CuPd nanocube. **d** An HAADF-STEM image with a zoomed-in view of the ordered CuPd nanocube. Blue and red balls in the overlaid model correspond to Cu and Pd atoms, respectively. **e** X-EDS elemental maps and the corresponding HAADF image of the CuPd nanocubes. **f** The XRD pattern of synthesized ordered CuPd nanocubes.

characterized by TEM to capture the morphological evolution of the ordered CuPd nanocubes (Supplementary Fig. 21). In the beginning, when the reaction temperature was raised to 250 °C for 1 min, only small spherical nanoparticles and some aggregates were observed. As the reaction time prolonged, nanocubes emerged. By the reaction time of 20 min, CuPd nanocubes were formed. The ICP results of the samples obtained at different reaction times suggest that Cu nanoparticles formed at the initial stage of the reaction and served as seeds for Pd to nucleate and grow (Supplementary Table 6). With the extension of reaction time, they grew into Cu-rich CuPd nanocubes. When the reaction was prolonged to 30 min, the ordered CuPd nanocubes formed. The final ordered CuPd structure with atom enrichment at the surface and corners could be attributed to the combination of galvanic replacement and Kirkendall effect[39,40]. This observation indicates that a multistage nucleation and growth process could promote the formation of ordered intermetallic structures, which was previously facilitated by tedious additional treatments (e.g., thermal annealing[18,41–43], electrochemical methods[34]).

**NO$_3$RR on ordered intermetallic CuPd nanocubes**. In order to evaluate the NO$_3$RR performance of the ordered CuPd nanocubes, these nanocubes were loaded onto carbon black (Vulcan XC-72R) and treated in acetic acid to remove the surfactants according to previously reported methods[44,45] (Supplementary Fig. 22). As a comparison, Cu nanocubes and Pd nanocubes were synthesized. As shown in Supplementary Figs. 23, 24, the size of Cu nanocubes is about 35 nm, and the size of Pd nanocubes is about 12 nm. The XRD patterns show that both of them are fcc structures (Supplementary Figs. 23b, 24b). Electrochemical measurements were performed using a three-electrode system in a gas-tight H-cell separated by an ion-exchange membrane (Nafion 117) at room temperature. Pt foil and Ag/AgCl (3.5 M KCl) were used as the counter and reference electrodes, respectively. The electrocatalysts were deposited onto 1 cm$^2$ carbon fiber paper, leading to a metal loading of ~0.2 mg cm$^{-2}$. The polarization curves were obtained by sweeping the potential from −0.5 to −1.7 V vs. Ag/AgCl at room temperature with a sweep rate of 20 mV s$^{-1}$ in the Ar-saturated 1 M KOH + 1 M KNO$_3$ solution at room temperature. As shown in

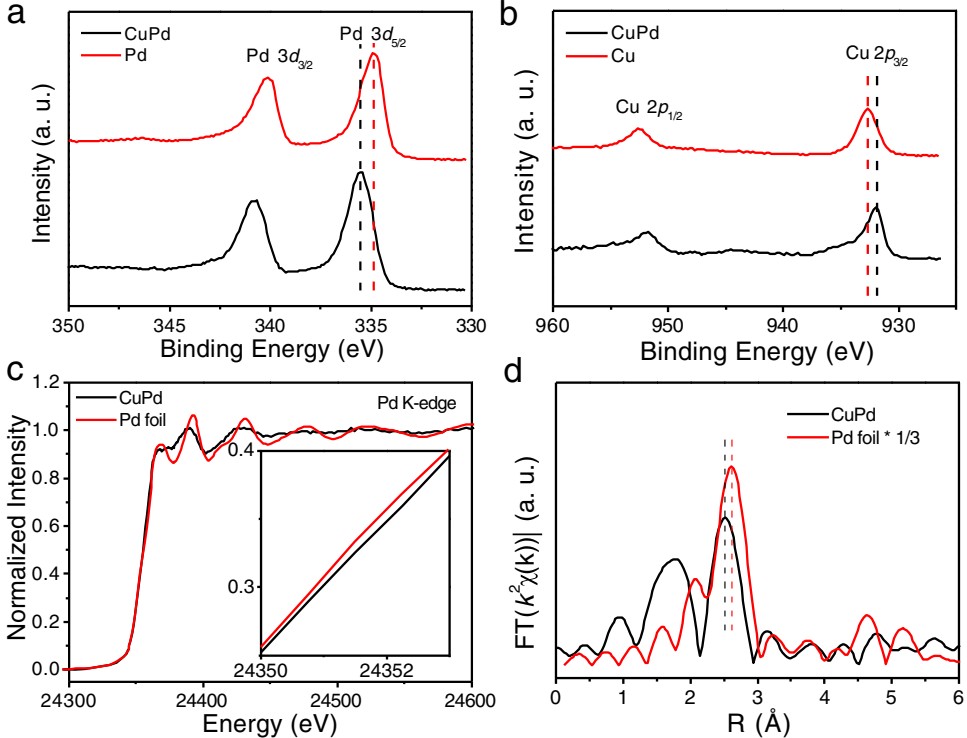

**Fig. 4 Structural characterizations of the ordered CuPd nanocubes. a** Pd 3d XPS spectra for ordered CuPd nanocubes and Pd nanocubes. **b** Cu 2p XPS spectrum for ordered CuPd nanocubes and Cu nanocubes. **c** Pd K-edge XANES spectra of ordered CuPd nanocubes and Pd foil reference, with a zoomed-in view of the Pd K-edge as inset. **d** EXAFS Fourier-transformed $k^2$-weighted $\chi(k)$ function spectra of ordered CuPd nanocubes and Pd foil reference.

Fig. 5a, the ordered CuPd nanocubes exhibit an onset potential of 0.1 V vs. RHE, which is much more positive than that of the Pd nanocubes (−0.05 V) and Cu nanocubes (−0.02 V). Figure 5b shows that the partial current densities of $NH_3$ on ordered CuPd nanocubes are much higher than that on Pd nanocubes and Cu nanocubes, suggesting that the ordered B2 CuPd nanocubes are much more active than Pd nanocubes and Cu nanocubes for the $NO_3RR$ toward $NH_3$. As shown in Supplementary Fig. 25, the ordered CuPd nanocubes show higher hydrogen evolution reaction (HER) activity than Cu nanocubes while Pd nanocubes perform best among all three tested. Cyclic voltammogram (CV) measurements at various scan rates (20, 40, and 60 mV s$^{-1}$, etc.) were conducted in static solution to estimate the double-layer capacitance ($C_{dl}$) by sweeping the potential across the non-faradaic region −0.1–0 V vs. Ag/AgCl (Supplementary Fig. 26). The $C_{dl}$ for the ordered CuPd nanocubes is calculated to be 3.59 mF cm$^{-2}$, which is smaller than that of Cu nanocubes (4.93 mF cm$^{-2}$) and Pd nanocubes (8.61 mF cm$^{-2}$), due to the larger size of the ordered CuPd nanocubes. The intrinsic activities for $NO_3RR$ on ordered CuPd, Cu, and Pd nanocubes are evaluated by normalizing catalytic currents to ECSAs (Supplementary Fig. 27). The ECSA-normalized partial current densities of $NH_3$ on ordered CuPd nanocubes are much higher than that on Cu nanocubes and Pd nanocubes, indicating the intrinsic activity for $NO_3RR$ on ordered CuPd nanocubes is superior to those on Cu nanocubes and Pd nanocubes, consistent with DFT-predicted activity trends in Supplementary Fig. 9.

Chronoamperometry (CA) measurements of catalysts were conducted at different potentials for 1 h in 1 M KOH + 1 M KNO$_3$ solution with continuous Ar bubbling at a rate of 20 standard cubic centimeters per minute (sccm) (Supplementary Fig. 28). The gas product was quantified by gas chromatography and only $H_2$ was identified from the competing HER. The colorimetric method using Nessler's reagent (Supplementary

Fig. 29) was employed to detect the quantity of produced $NH_3$ and ion chromatography was employed to detect the quantity of produced $NO_2^-$ (Supplementary Fig. 30). To avoid the loss of products from continuous Ar flow, a glass vial filled with 0.1 M HCl was set at the end of the outlet tube as the trap. The total $NH_3$ production yield was the summation of $NH_3$ in the electrolyte and 0.1 M HCl. The FEs and $NH_3$ yield rates of the electrocatalysts are shown in Fig. 5c, d. The ordered CuPd nanocubes demonstrated high selectivity toward $NH_3$ production from $NO_3RR$ with a FE of 92.5% at −0.5 V vs. RHE and a high yield rate of 6.25 mol h$^{-1}$ g$^{-1}$ at −0.6 V vs. RHE, outperforming most of the reported catalysts (Supplementary Table 7)[6,46–48]. The main byproduct of $NO_3RR$ on ordered CuPd nanocubes is $NO_2^-$, as detected and quantified by ion chromatography. The FE of $NO_2^-$ starts from as high as 20.7% at −0.2 V vs. RHE, followed by a significant decrease to a minimal of ~2.67% at −0.6 V vs. RHE. This suggested that $NO_2^-$ could be an intermediate product and can be further reduced to $NH_3$ at a more negative potential, which is consistent with our theoretical results. Furthermore, in situ attenuated total reflectance surface-enhanced infrared absorption spectroscopy (ATR-SEIRAS) measurements were carried out to identify the intermediates of the $NO_3RR$ process. Supplementary Figure 31 shows the ATR-SEIRAS spectra results collected from ordered CuPd nanocubes during a CV cycle between 0.965 V and −0.835 V vs. RHE at 5 mV/s in 0.1 M KOH solution (Supplementary Fig. 31a) and 0.1 M KOH + 1 M KNO$_3$ solution (Supplementary Fig. 31b). According to previous reports, the absorption at 1645 cm$^{-1}$ can be attributed to the H–O–H bending of water molecules[49]. The absorption at around 1365 cm$^{-1}$ in Supplementary Fig. 31b is due to the adsorption of nitrate ions[50]. The intensity of the peak increased as the potential became more negative. There is no N=N stretching band at ~2010 cm$^{-1}$ appeared, which indicated that $N_2H_x$ is not a reaction intermediate of the nitrate reduction on ordered CuPd nanocubes[51].

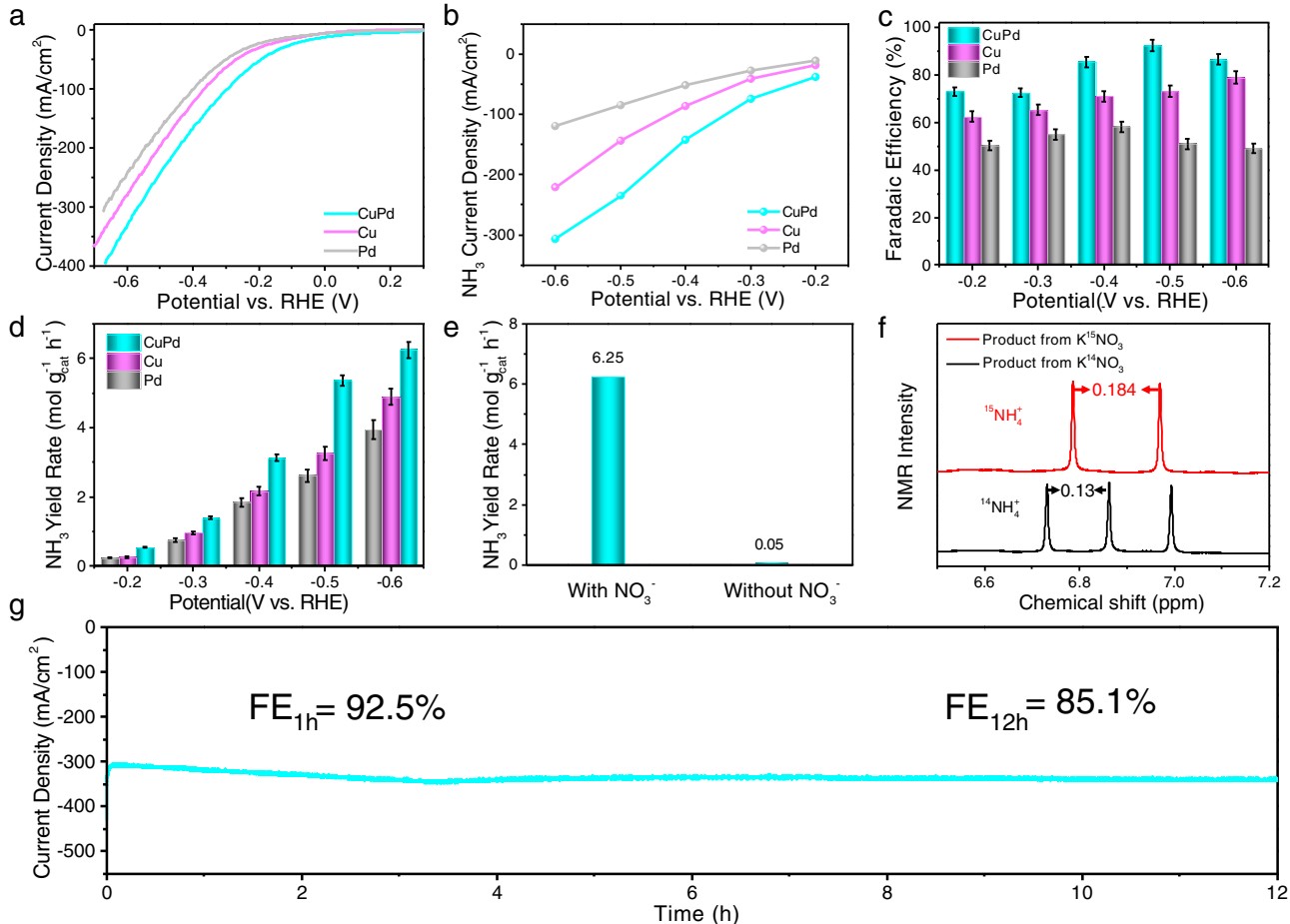

**Fig. 5 Electrocatalytic NO₃RR performance of the ordered CuPd nanocubes. a** Linear scan voltammetry curves of ordered CuPd nanocubes, Cu nanocubes, and Pd nanocubes normalized to the geometric area. **b** Partial $NH_3$ current densities normalized to the geometric area. **c** FE of $NH_3$ at different potentials, The error bars correspond to the standard deviation from three independent measurements. **d** $NH_3$ yield rate of ordered CuPd nanocubes, Cu nanocubes, and Pd nanocubes at various potentials. **e** The yield rate of $NH_3$ with or without nitrate at $-0.6$ V vs. RHE. **f** NMR spectrum of the products generated during the electrocatalytic NO₃RR with ordered CuPd nanocubes in 1 M $K^{15}NO_3$ or 1 M $K^{14}NO_3$ at $-0.5$ V vs. RHE. **g** Stability test by running the CA measurement on the ordered CuPd nanocubes at $-0.5$ V vs. RHE for 12 h.

To confirm the NO₃RR performance, the control experiment was performed at $-0.6$ V vs. RHE for 1 h in 1 M KOH solution without $KNO_3$. As shown in Fig. 5e, there is almost no $NH_3$ detected in the electrolyte. To confirm the produced $NH_3$ originated from the feeding nitrate solution, $^{15}N$ isotope labeling experiments were conducted. After electrolysis at $-0.5$ V vs. RHE for 1 h in 1 M $K^{15}NO_3$, no characteristic triple coupling peaks of $^{14}NH_4^+$ could be detected in the $^1H$ nuclear magnetic resonance ($^1H$ NMR) spectra of the electrolyte, while only doublet peaks representing $^{15}NH_4^+$ were observed (Fig. 5f), indicating that the produced $NH_3$ entirely comes from the electroreduction of nitrate. To evaluate the long-term stability for future practical applications, we performed the durability test by running the chronoamperometry on the ordered CuPd nanocubes at $-0.5$ V vs. RHE for 12 h. As shown in Fig. 5g, the current density exhibits no appreciable decrease over 12 h of continuous operation, and the overall 12h FE of $NH_3$ is ~85.1%. There is a slight decrease in the NO₃RR electrocatalytic performance, possibly due to a small fraction of the catalyst falling off from the carbon paper. After the long-term electrolysis, the catalyst was removed from carbon paper and characterized by TEM and XPS. As shown in Supplementary Fig. 32, the cubic morphology is maintained, and there is no apparent aggregation observed. The HRTEM image (Supplementary Fig. 32c) shows clear lattice fringes with a

lattice spacing of 2.95 Å, which can be assigned to the (100) planes of the ordered CuPd intermetallic phase. Moreover, the ICP-MS result shows that the Pd/Cu molar ratio is still close to 1:1. We also performed XRD measurement after the stability test (Supplementary Fig. 33), which showed that the CuPd nanocubes still maintain the ordered B2 phase after long-term electrolysis. Furthermore, XPS analysis reveals that no obvious chemical state changes after the stability test (Supplementary Fig. 34). These results illustrate the exceptional chemical and structural stability of the ordered CuPd nanocubes. Besides, electrochemical impedance spectroscopy (EIS) was recorded on these electro-catalysts to provide further insight into electrode kinetics (Supplementary Fig. 35). Representative Nyquist plots show that the ordered CuPd nanocubes have a smaller charge transfer resistance ($R_{ct}$) than that of Cu nanocubes, indicating the fast Faradaic process and thus superior NO₃RR kinetics for the ordered CuPd nanocubes, which stresses the importance of Pd in the intermetallic structure that modifies the electric conductivity of the ordered CuPd nanocubes.

Besides, in order to further validate the theoretical prediction of reactivity trends on pure metals in Fig. 1c, Cu spherical nanoparticles and Au particles (primary nanocubes) were synthesized (Supplementary Fig. 36). As shown in Supplementary Fig. 37, Cu nanocubes with (100) facets show higher partial $NH_3$

current density, FE and $NH_3$ yield rate than those of Cu nanoparticles, which indicates that Cu(100) has higher electrocatalytic activity for $NO_3RR$ to $NH_3$ than Cu(111) as DFT-predicted. The activities of $NO_3RR$ to $NH_3$ on nanocubes with (100) facets follow the trend Cu > Pd > Au, which is consistent with DFT-predicted activity trends in Fig. 1c.

## Discussion

With electrocatalytic $NO_3^-$ reduction to $NH_3$ on metal nanocatalysts as an example, we have demonstrated that interpretable ML of ab initio adsorption properties provides physical insights into the nature of chemical bonding that can be leveraged to break linear adsorption-energy scaling limitations of catalytic performance. Bayeschem models of two reactivity descriptors, i.e., bridge-bidentate *$NO_3$ and hollow *N, suggest that both adsorbates behave similarly in orbital hybridization upon a perturbation of the local electronic structure, e.g., $d$-band center. However, *N exhibits a more prevalent repulsion contribution on (100)-like sites than (111) as increasing interatomic coupling strengths, while the bridge-bidentate *$NO_3$ is not directly influenced due to large distances. These machine-learned insights can be leveraged for breaking linear scaling relationships, specifically at (100)-terminated surface sites of B2 intermetallics, in which the layer separation is small compared to other site motifs of ordered intermetallics. A handful of B2 systems are predicted to be close to the top of the activity volcano plot because of a weakened *N binding and enhanced *$NO_3$ adsorption than Cu(100). Among those predicted, our synthesis strategy has enabled the direct, solution-phase formation of ordered intermetallic CuPd nanocubes that demonstrated highly efficient $NO_3RR$. This study showcases a strategy for breaking the linear scaling relations on ordered intermetallic catalysts by harnessing the Pauli repulsion of the metal $d$-states with adsorbate frontier orbitals. Moreover, it highlights the benefit of interpretable ML and DFT calculations together with the synthesis of structurally controlled well-defined nanocrystals in suggesting and verifying the governing physical insights, providing a methodological basis for fine-tuning of electrocatalysts for improved efficiency.

## Methods

**Chemicals**. Palladium(II) chloride ($PdCl_2$, 99.9%), sodium tetrachloropalladate ($Na_2PdCl_4$, 98%), copper(II) acetylacetonate [$Cu(acac)_2$, 99.9%], copper bromide (CuBr, 98%), oleylamine (OAm, 70%), trioctylphosphine (TOP, 97%), trioctylphosphine oxide (TOPO, 99%), poly(vinyl pyrrolidone) (PVP), L-ascorbic acid (AA), 1,2-tetradecanediol (TDD), potassium bromide (KBr), potassium hydroxide (KOH), potassium nitrate ($KNO_3$ and $K^{15}NO_3$), acetic acid, sulfuric acid ($H_2SO_4$), hydrogen peroxide ($H_2O_2$), Nessler's reagent, and Nafion (5 wt %) were all purchased from Sigma–Aldrich. Hexane and ethanol were technical grade and used without further purification.

**Synthesis of ordered intermetallic CuPd nanocubes**. In a modified procedure[52], 0.1 mmol $PdCl_2$, 0.1 mmol $Cu(acac)_2$, 0.4 mmol TDD, and 10 mL OAm were added into a 25 mL three-necked flask under stirring. The mixture was heated at 80 °C for 30 min under $N_2$ atmosphere. Then 0.5 mL TOP was injected into the solution. After that the mixture solution was heated to 250 °C and reacted for 30 min, generating a black solution. The black precipitate was cooled down to room temperature, washed three times with excessive ethanol, and dispersed in hexane.

**Synthesis of Cu nanocubes**. In a modified procedure[53], 0.2 mmol of CuBr, 0.5 mmol TOPO, and 10 mL OAm were loaded into a 25 mL three-necked flask under stirring. The mixture was heated at 80 °C for 30 min under $N_2$ atmosphere. Then the mixture was further heated to 250 °C and reacted for 30 min, generating a reddish solution. The precipitate was cooled down to room temperature, washed three times with excessive ethanol, and dispersed in hexane.

**Synthesis of Pd nanocubes**. In a modified procedure[54], 105 mg PVP, 60 mg AA, 300 mg KBr, and 8 mL deionized water were added into a 25 mL vial under stirring. The mixture was heated at 80 °C for 10 min. Then 3 mL of an aqueous solution of $Na_2PdCl_4$ (57 mg) was injected into the reaction vessel with a pipette. After that, the solution was kept at 80 °C for 3 h, generating a black solution. The black

precipitate was cooled down to room temperature, washed five times with water and ethanol, and dispersed in ethanol.

**Preparation of carbon-supported catalysts (20% loading)**. To prepare carbon-supported catalysts, the catalysts were loaded onto carbon black (Vulcan XC-72R) according to previously reported methods[44,45]. The hexane dispersion of 10 mg of intermetallic CuPd nanocubes was mixed with 40 mg carbon black and sonicated for 2 h. The product was collected by centrifugation. Afterwards, the catalysts were immersed in a mixture of 10 mL ethanol and 10 mL acetic acid for 10 h at 70 °C to remove organic ligands on the surface of these CuPd nanocubes. The catalysts were washed three times with excessive ethanol and dried for 8 h in a vacuum oven at 60 °C. The control samples of Cu and Pd nanocubes were loaded onto carbon via a similar approach.

**Characterizations**. XRD was performed on a Philips X' Pert PRO SUPER with Cu Kα ($\lambda = 1.54056$ Å). XPS was performed on a PHI Versa Probe III scanning XPS microscope using a monochromatic Al K-alpha X-ray source (1486.6 eV). The sample's morphology was characterized by SEM (Zeiss Supra 40) and TEM (EM-420). HRTEM, HAADF-STEM, X-EDS, and EDS mapping were conducted on a JEOL ARM 200CF equipped with an Oxford Instrument X-ray Energy Dispersive Spectrometer. The element contents of the products were determined by ICP-OES on a SPECTRO GENESIS ICP spectrometer. The gas product was quantified by gas chromatography (Agilent 7890B). The colorimetric method with Nessler's reagent on a UV-vis spectrophotometer (Agilent 3500) and ion chromatography (Metrohm Eco IC) was used to quantify the produced ammonia. Ion chromatography was also used to quantify the produced nitrite. The $^1H$ NMR signal was recorded on a Bruker 400 MHz system. The X-ray absorption spectra of Pd and Cu K-edges were obtained at the beamline 12-BM-B station of the Advanced Photon Source at Argonne National Laboratory. Both Pd and Cu K-edge XANES and EXAFS were measured under fluorescence mode by a Vortex ME4 detector. All XAS data analyses were performed with the Athena software package to extract XANES and EXAFS. Fourier-transform infrared spectroscopy (FTIR) was performed on an Agilent Cary 630.

**Electrochemical measurements**. Electrochemical measurements were carried out on a BioLogic electrochemical workstation. All measurements were performed in a gas-tight H-cell using a three-electrode system and an ion-exchange membrane (Nafion 117) at room temperature. Before testing, the Nafion 117 membrane was immersed in 5% $H_2O_2$ solution at 80 °C for 1 h, then in 0.5 M $H_2SO_4$ solution at 80 °C for an additional hour, and finally washed with deionized water. The electrode preparation and electrochemical measurements are done using the previously reported procedure[55]. An Ag/AgCl electrode (3.5 M KCl) and a Pt foil were used as the reference and counter electrodes, respectively. The potentials were measured against the Ag/AgCl electrode and converted to RHE according to E (vs. RHE) = E (vs. Ag/AgCl) + 0.198 V + 0.059 × pH. The working electrodes were prepared as follows: 5 mg of carbon-supported catalyst, 1 ml isopropanol, and 20 μl of Nafion solution (5 wt%, Sigma–Aldrich) were mixed and ultrasonicated for more than 30 min to generate a homogeneous ink. Then, 200 μl of the catalyst ink was deposited onto a 1 cm$^2$ carbon fiber paper by drop-casting, resulting in a metal loading of ~0.2 mg cm$^{-2}$. Before the electrochemical measurement, the 1 M KOH + 1 M $KNO_3$ electrolyte was purged with Ar for at least 30 min. The LSV curves were obtained by scanning the potential from −0.5 to −1.7 V vs. Ag/AgCl at a rate of 20 mV s$^{-1}$. CA measurements were conducted at various potentials in 1 M KOH + 1 M $KNO_3$ solution with an Ar flow rate of 20 sccm in the cathodic compartment. CV measurements at various scan rates were performed to estimate the $C_{dl}$ of the catalysts in a static solution by scanning the potential between −0.1 and 0 V vs. Ag/AgCl. EIS measurements were carried out in the potentiostatic mode at −1.2 V vs. Ag/AgCl, applying a 5 mV AC dither and scanning from 100 kHz to 100 MHz.

**Quantification of products**. The gas product was quantified by gas chromatography (Agilent 7890B). The quantity of ammonia produced was measured using a colorimetric method with Nessler's reagent. All test solutions were incubated under dark conditions at room temperature for 20 min before UV-vis tests. The absorbance at 420 nm for each solution was measured with a UV-vis spectrophotometer (Agilent 3500). A series of reference solutions with suitable $NH_4Cl$ concentrations was created to plot a calibration curve. Electrolytes after catalysis were diluted to ensure the ammonia concentrations in the test solutions were in the linear range of Nessler's method. The concentrations of ammonia in the electrolytes were obtained with this as-obtained calibration curve. Ion chromatography was also used to detect the liquid products. The liquid products from the isotopic experiment were analyzed by $^1H$ NMR using dimethyl sulfoxide (DMSO) (20%) and 1 mM maleic acid as the internal standard. The pH of the after-reaction electrolyte was adjusted to 2 using 1 M HCl.

Faradaic efficiency was calculated according to the following equation:

$$FE = (n \times C \times V \times F)/(i \times t) \qquad (1)$$

Yield rate was calculated according to the following equation:

$$r = (C \times V)/(t \times m) \tag{2}$$

where n is the number of electron transfers towards the formation of 1 mol product; C is the concentration of product (M); V is the volume of catholyte (mL); F is the Faraday constant (96,485 $C \cdot mol^{-1}$); i is the reduction current; t is the total reaction time, and m is the catalyst mass.

**DFT calculations**. DFT calculations were performed using VASP[56,57] and Quantum ESPRESSO (QE)[58] at the GGA level using the RPBE[59] functional. QE calculations were used to get the *N and *NO_3 adsorption energies and the atom and molecular projected density of states used to optimize the Bayeschem models. For QE calculations, the core electrons were treated using ultrasoft pseudopotentials with kinetic energy cutoffs of 500 eV for wavefunction and 5000 eV for charge density. In order to speed up calculations, partial occupancies were set using the Fermi-Dirac smearing with a smearing parameter of 0.1 eV. For all the other electronic structure calculations, VASP with projector augmented wave pseudopotentials was used. A planewave energy cutoff of 450 eV was used and the same k-point setting as QE were used for VASP. The Methfessel-Paxton smearing scheme was used with a smearing parameter of 0.2 eV for adsorbate systems and 0.001 eV for molecules. Electronic energies are extrapolated to $k_B T = 0$ eV. For both VASP and QE the same model systems were used. The (100) surfaces were modeled using a $3 \times 3 \times 6$ supercell with the bottom 4 layers fixed, and the (111) surfaces were modeled using a $4 \times 4 \times 4$ supercell with the bottom two layers fixed. The Brillouin zones for (111) and (100) surfaces were sampled using the Monkhorst-Pack meshes of $3 \times 3 \times 1$ and $4 \times 4 \times 1$, respectively. A vacuum of 15 Å was used between two periodic images. Geometries were optimized until the maximum forces was less than 0.05 eV/Å. Calculations for molecules or atoms were done in a $15 \times 15 \times 15$ Å box using the Gamma point. Atom and molecular projected density of states[60] were calculated in QE on a denser k-point sampling of $12 \times 12 \times 1$ with an energy spacing of 0.01 eV.

To convert the DFT electronic energies to free energies, we added the zero-point energy (ZPE), heat capacity, and entropic contributions. ZPE and entropic corrections were calculated within the harmonic oscillator approximation. For adsorbates, all degrees of freedom are treated as vibrational. Using the Schumann et al.[61] approach, frequencies less than 100 $cm^{-1}$ are treated as pseudo-translational/rotational and thus replaced by 12 $cm^{-1}$. For gas-phase species, translational and rotational contributions to the internal energy and entropy are considered using statistical thermodynamics. All the corrections are calculated at 298 K and tabulated in Table S1. The free energy of liquid water is calculated using the gas phase at its vapor pressure at 298 K (0.035 bar). The free energy of an adsorbed system *A at the gas-solid interface is given by the following equation:

$$G_{*A} = E_{*A}^{DFT} + ZPE - TS + \int_0^{298} C_p dT. \tag{3}$$

At the solid-electrolyte interface under constant potential conditions, the Gibbs free energy of an adsorbed system *A is then calculated as

$$G_{*A} = E_{*A}^{GC-DFT} + ZPE - TS + \int_0^{298} C_p dT - \mu_e \times N_e, \tag{4}$$

where $\mu_e$ is the chemical potential of the electron (same as the Fermi level since the vacuum potential is 0). $N_e$ is the number of electrons added (positive in sign) or removed (negative in sign) relative to the total number of electrons in the charge-neutral system. In this case, $E_{*A}^{GC-DFT}$ is the electronic energy calculated from the GC-DFT. Grand-canonical DFT calculations (GC-DFT) were performed in order to simulate the solid-electrolyte interfaces with the applied bias and solvation. In GC-DFT calculations, the applied potential bias is controlled by changing the number of electrons in the simulation cell, which modifies the Fermi level relative to the vacuum potential and thus the work function of the system ($\epsilon_F = -\phi$). The equation below shows how the work function ($\phi$) is connected to the applied potential at the SHE (standard hydrogen electrode) scale.

$$U_{SHE} = \frac{\phi - 4.43}{e} \tag{5}$$

Therefore, systems can be simulated at a specific potential by optimizing the number of electrons to reach the target work function. Solvation was included through the VASPsol[62,63] with a continuum dielectric description of electrolytes. The surface tension parameter (0), the dielectric constant (78.4), and the Debye screening length (3 Å) were set in all GC-DFT calculations[64,65]. The electrode potential correction to the free formation energy of a surface intermediate was obtained from GC-DFT calculations on Cu(100) at 0 V vs RHE. This correction is assumed to be constant, which is a reasonable approximation across metal surfaces at this potential (Supplementary Table 2). Examples of calculating adsorbate free formation energies are shown in the Supplementary discussion and thermodynamic cycle in Supplementary Fig. 11 for nitrate adsorption. We include the most stable adsorbate configurations for pure metals in a Github repository (https://github.com/hlxin/nitraterr).

**Bayesian theory of chemisorption (Bayeschem)**. Within the d-band theory of chemisorption, the adsorption energy ($\Delta E_A$) of a species at metal surfaces can be

calculated as:

$$\Delta E_A = \Delta E_0 + \Delta E_d \tag{6}$$

where $\Delta E_0$ is the energy change due to the interaction between the adsorbate and the free-electron-like sp-states of the transition metals, while the $\Delta E_d$ is due to the interaction between the adsorbate and the localized d-states. It is assumed that the sp-band contribution is constant for a given surface facet due to similar sp-states of the transition metals. Whereas the variations in binding energies are explained due to differences in the d-contribution[66]. The d-contribution can be further decomposed into repulsive orbital orthogonalization and attractive orbital hybridization.

$$\Delta E_d = \Delta E_d^{hyb} + \Delta E_d^{orth} \tag{7}$$

The d-hybridization contribution can be calculated with the Newns-Anderson model, which depends on several parameters ($\alpha, \beta, \Delta_0, \epsilon_a$), as shown below for a simple case with one valence state of the adsorbate.

$$\Delta E_d^{hyb} = \frac{2}{\pi} \int_{-\infty}^{\epsilon_f} \tan^{-1} \frac{\Delta(\epsilon)}{\epsilon - \epsilon_a - \Lambda(\epsilon)} d\epsilon - \frac{2}{\pi} \int_{-\infty}^{\epsilon_f} \tan^{-1} \frac{\Delta_0(\epsilon)}{\epsilon - \epsilon_a} d\epsilon \tag{8}$$

where the chemisorption function $\Delta(\epsilon)$ depends on the surface density of states ($\rho_d$):

$$\Delta(\epsilon) = \Delta_0 + \pi \beta V_{ad}^2 \rho_d \tag{9}$$

The repulsive contribution from each adsorbate frontier orbital is derived by treating the adsorbate-substrate interaction as a two-level system.

$$\Delta E_d^{orth} = 2(\tilde{n}_a + f)\alpha\beta V_{ad}^2, \tag{10}$$

in which f and $\tilde{n}_a$ are the filling of the metal d-states and the adsorbate resonance state. A Bayesian learning approach was used to optimize the model parameters mentioned above by learning from ab initio adsorption properties. Further details on this approach can be seen in ref. [15]. Bayesian models were optimized for *NO_3 and *N on the fcc(100) and fcc(111) metal surfaces with the parameters shown in Supplementary Figs. 2–5.

## Data availability
The data that support the plots within this paper and other findings of this study are available from the corresponding author upon reasonable request.

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

## Acknowledgements

We acknowledge the funding support from the U.S. National Science Foundation (NSF) (CHE-2102363). H.Z. acknowldges the NSF CAREER program (CBET-2143710). H.S.P., Q.M., and H.X. acknowledge the NSF CAREER program (CBET-1845531). The computational resource used in this work is provided by the advanced research computing at Virginia Polytechnic Institute and State University. Q.H. would like to acknowledge the support by National Research Foundation (NRF) Singapore, under its NRF Fellowship

(NRF-NRFF11-2019-0002). This research used resources of the Advanced Photon Source, a U.S. Department of Energy (DOE) Office of Science User Facility operated for the DOE Office of Science by Argonne National Laboratory under Contract No. DE-AC02-06CH11357. We would like to thank Prof. Sen Zhang and his student Grayson Johnson from the University of Virginia for their help with ATR-SEIRAS.

## Author contributions

Q.G., H.Z., and H.X. conceptualized the project. H.Z. and H.X. supervised the project. Q.G. planned and performed the catalyst synthesis, conducted the electrocatalytic tests, collected, and analyzed the data. H.S.P., Y.H., and Q.M. did the theoretical calculations. S.L. and Q.H. collected and analyzed the STEM data. H.Z. performed the synchrotron XAS measurements. X.H. and Z.Y. helped with synthesis of the catalysts and collected the data. Q.G., H.S.P., H.Z., and H.X. wrote the manuscript. All authors discussed the results and commented on the manuscript.

## Competing interests

The authors declare no competing interests.
