## [Peer review file · Nature Communications]

REVIEWER COMMENTS

Reviewer #1 (Remarks to the Author):

This work reports a combined computational and experimental study that uses interpretable machine learning to predict and then synthesize new catalysts for the electrocatalytic reduction of nitrate to ammonia. The key insight provided in this work is the prediction of specific intermetallic catalyst motifs that can break the scaling relationship between N^* and NO_3^* , thus allowing certain catalyst compositions to approach the top of the activity volcano for this reaction. Interpretable machine learning provides deep electronic-level insight into how this is achieved. Namely, (100)-like sites on B2 intermetallics can take advantage of differences in orbital overlaps experienced by NO_3^* at the bridge site and by N^* at the hollow site. Using this insight, the authors determine that a CuPd intermetallic ought to perform better than pure Cu, which is then demonstrated in the experimental results.

This is a very nice study that will appeal to a broad audience, as it incorporates elements of data science, computational chemistry, and experiment all to study an important application (i.e., nitrate reduction). That being said, in my opinion there are several ways that the manuscript can be clarified and strengthened. For this reason, I recommend that this work be accepted for publication if the following points can be adequately addressed.

1. A key finding reported in the manuscript is the prediction that CuPd will outperform Cu by breaking the scaling relationship and approaching the top of the activity volcano. This prediction is made based on data in Fig. 2. However, it is difficult to see using just the figure exactly what the predicted difference is in activity given the shift in ΔG_{max} going from Cu to CuPd. Could the authors clarify the numerical values for the ML-predicted and DFT-calculated ΔG_{max} for Cu versus CuPd? I believe this information is also presented in S. Figure 8, but again it is difficult to discern the difference between Cu and CuPd. I am asking for these clarifications because it is difficult to judge how clearly the methodology distinguishes the predicted activity of Cu from that of CuPd. Does it come down to a difference in ΔG_{max} on the order of ~ 0.2 eV, and if so how does this difference compare to the expected error range of the methodology?
2. Given the results predicting that CuPd will be near to the top of the activity volcano, I was expecting the authors to report a deeper analysis of the electronic structure of CuPd as it relates to N^* and NO_3^* binding. However, the authors only confirm their prediction with a DFT reaction energy diagram in S. Figure 8. The manuscript would be greatly enhanced if the authors were to include a much deeper analysis of the CuPd system at the electronic level (presented side by side with pure Cu for comparison), which will clearly illustrate how NO_3^* and N^* adsorption are independently tuned by the underlying electronic structure.

3. On Pg. 3 the authors state “Single-crystal experiments showed that NO₃RR on Cu is structure sensitive with (100)-oriented surface sites more active toward NH₃ formation than the (111) counterparts.” I believe this sentence should refer to reference 5 (Koper and co-workers) and not reference 6.

4. On Pg. 10 the authors state “DFT calculations of reaction pathways confirmed the model prediction with CuPd limited by *NO₃ adsorption and Cu termination is more stable than Pd termination (Supplementary Fig. 8).” I don’t see anything in this figure regarding the stability of the Cu termination versus the Pd termination. How was this stability difference determined?

5. On Pg. 6 the authors state “Moreover, *NO₂ formation from NO₂⁻ is more exergonic on Cu(111) than that on Cu(100), resulting in a slower further reduction of NO₂⁻ on (111)-type sites as observed.” I do not follow why this would lead to slower reduction of NO₂⁻ on (111) sites. Please clarify the logic here. Did the authors mean to say that NO₂⁻ formation from NO₂* is more exergonic on Cu(111), and thus NO₂⁻ desorbs more readily from (111) and therefore reduction beyond NO₂⁻ occurs less readily on (111)? If so, it is interesting to note that a similar effect was reported for Pd(111) versus Pd(100) in Hatzell et al. ACS Catal. 2021, 11, 12, 7568–7577.

6. In the SI on Ln. 72 the authors state “An additional intermediate dependent constant was added to account for the electrolyte/electrode interface and is tabulated in the last column of Supplementary Table 1.” Can the authors clarify how this correction was determined? It appears that the authors use GC-DFT on Cu(100) to derive this correction value, and then assume that the same value can be used for all surfaces. If this is the case, can the authors include benchmarking tests on an additional surface (or two) to quantify error associated with such an assumption?

7. In the SI on Ln. 52 the authors state “To get the Gibbs free energy of the alkaline reaction at a given pH, an additional correction $0.059 \times \text{pH}$ was added (+1.22 eV at pH 14 in our system).” I suggest that the authors re-word this and state that the correction at pH 14 is 0.83 eV, which summed with the value in the preceding sentence of 0.39 eV equals a total correction of +1.22 eV.

Reviewer #2 (Remarks to the Author):

This work proposes ordered B2 CuPd nanocubes that demonstrate exceedingly high performance for NO₃RR to ammonia with a FE of 92.5% at -0.5 VRHE and a yield rate of 6.25 mol h⁻¹ g⁻¹ at -0.6 VRHE. This study provides machine-learned design rules that can be realized on (100)-type sites of ordered B2 intermetallics, in which the orbital overlap between the hollow *N and subsurface metal atoms is significant while the bridge-bidentate *NO₃ is not affected. However, some important discussion and

information are missing such as the investigation about the origin of activities are insufficient. I recommend reconsidering this work after a major revision. Please see the comments below.

Major comments

1. The author insisted that the conventional method for synthesis of intermetallic structure has a disadvantage for sintering or preserving the nanocrystals, but the solution-phase synthesis author used, the NP size is already large over 20 nm and the low-resolution TEM (Supplementary Figure 26) showed the nanocrystal shapes are collapsed. It's hard to be sure what the benefits of this method are.
2. The applied potential -0.5 VRHE is quite negative and low. This potential range is enough to produce much hydrogen, but authors provide 92.5% FE for ammonia. Pd is also well-known material for high N₂ selectivity, but there is no information about N₂ FE. Also, more negative potentials should produce more hydrogen but why the hydrogen FE is getting lower in Supplementary figure 23b, d, f?
3. Please provide B2 intermetallic CuPd/C catalysts pristine of TEM information. Supplementary Figure 26 showed only after stability test TEM data, no pristine data for carbon supported CuPd catalysts. Then readers can compare the stability test data. Also, it is hard to verify the intermetallic structure of CuPd nanocrystals is remained because of very low-resolution. But it seems the shapes are collapsed. Please provide the high-resolution TEM image after the test to prove the intermetallic structure remains. Not only this, change of chemical compositions, states (XPS for Supplementary Figure 27b has a poor quality for discussing), leaching concentrations are missing to discuss the stability. After the test, the lower FE of 85.1% is measured. Please provide the discussion for this.
4. Please provide HER LSV curves of all catalysts in Figure 5a for reference.
5. Please provide the quantification information for NO₃⁻ and NO₂⁻ with a curve of UV-vis spectra and wavelength.
6. Please provide Cu-O peak reference information for XAS data in Fig 4c,d. Also, the authors need to provide more discussion for changing of R difference in Fig. 4d.

Reviewer #3 (Remarks to the Author):

Summary

The authors investigate the electrochemical nitrate reduction reaction (NO₃RR) to ammonia theoretically and experimentally. DFT calculations demonstrate that the reactivity and selectivity on the pure metal surface are constrained by the scaling between *NO₃ and *N adsorption energies. As the 2 species adsorb via different modes on the surface, the Bayesian theory of chemisorption attempts to find catalytic systems that affect the binding energies of the 2 adsorption modes differently to break this scaling relation. B2 ordered intermetallics were found to be both active and selective because these catalysts can break the scaling between *NO₃ and *N adsorption strength through the site-specific Pauli repulsion of the metal d-states with adsorbate frontier orbitals. Guided by the theoretical prediction, the B2 ordered CuPd nanocubes are synthesized, which exhibit outstanding catalytic performance for NO₃RR to ammonia. This is an interesting study and demonstrates the power of the machine learning approach for finding active intermetallics with multiple functionalities and definitely excludes the use of more time-consuming combinatorial techniques for catalyst evaluation. The manuscript is well written and suitable for the standards of Nature communication after some revisions.

Major comments

1. The discussion about the reaction mechanism is insufficient. Are there other reaction intermediates at play? Are the most stable adsorption geometries of intermediates on all pure metals and intermetallics identical? Is the proposed reaction mechanism (Figure 1b) the most favorable pathway? Can the reaction proceed through other mechanisms, for instance, NO₃ → NO₂ → NO → NOH → N → NH → NH₂ → NH₃?
2. The thermodynamic calculations are not enough to capture the activity trends (Figure 1c) due to the lack of energy barriers and reaction rates. At least, the author should provide enough justification for not sampling transition states and for an impact publication, barriers would be better to report.
3. The authors should present more justification for selecting the binding energy of *N as a descriptor based on its role in the reaction mechanism.
4. The authors need to show relevant experimental evidence on other metal catalysts, besides Cu(100) and Cu(111), to validate the theoretical prediction of reactivity on pure metals (Figure 1c).
5. It was reported that the CuPd nanocubes are enriched with the Cu phase at the outer surfaces/edges. The authors should validate this through a more direct physical or chemical characterization technique.

Minor comments

1. On the 13th line of Page 10, Fig.2f should be corrected into Fig.2g
2. The authors should supplement the free energy profile of NO₃⁻ reduction to NH₃(g) on Pd(100) and compare it with those on CuPd(100) and Cu(100)

Incl: Our answers to reviewer's questions

Reviewer #1 (Remarks to the Author):

This work reports a combined computational and experimental study that uses interpretable machine learning to predict and then synthesize new catalysts for the electrocatalytic reduction of nitrate to ammonia. The key insight provided in this work is the prediction of specific intermetallic catalyst motifs that can break the scaling relationship between N^* and NO_3^* , thus allowing certain catalyst compositions to approach the top of the activity volcano for this reaction. Interpretable machine learning provides deep electronic-level insight into how this is achieved. Namely, (100)-like sites on B2 intermetallics can take advantage of differences in orbital overlaps experienced by NO_3^* at the bridge site and by N^* at the hollow site. Using this insight, the authors determine that a CuPd intermetallic ought to perform better than pure Cu, which is then demonstrated in the experimental results.

This is a very nice study that will appeal to a broad audience, as it incorporates elements of data science, computational chemistry, and experiment all to study an important application (i.e., nitrate reduction). That being said, in my opinion there are several ways that the manuscript can be clarified and strengthened. For this reason, I recommend that this work be accepted for publication if the following points can be adequately addressed.

Response: We are very grateful for the reviewer's positive feedback on this work. We thank the reviewer for his/her recommendation and constructive comments. We have addressed the raised comments point by point as follows:

Q1. A key finding reported in the manuscript is the prediction that CuPd will outperform Cu by breaking the scaling relationship and approaching the top of the activity volcano. This prediction is made based on data in Fig. 2. However, it is difficult to see using just the figure exactly what the predicted difference is in activity given the shift in ΔG_{max} going from Cu to CuPd. Could the authors clarify the numerical values for the ML-predicted and DFT-calculated ΔG_{max} for Cu versus CuPd? I believe this information is also presented in S. Figure 8, but again it is difficult to discern the difference between Cu and CuPd. I am asking for these clarifications because it is difficult to judge how clearly the methodology distinguishes the predicted activity of Cu from that of CuPd. Does it come down to a difference in ΔG_{max} on the order of ~ 0.2 eV, and if so how does this difference compare to the expected error range of the methodology?

Response: We thank the reviewer for these questions and want to clarify how machine learning (ML) was used in our design approach. *Bayeschem* ML models were employed to extract physical insights into chemical bonding of $*NO_3$ and $*N$ species at metal sites. In Figure 2, we independently tune the site electronic structure and the adsorbate-metal coupling strength for Cu(100) and Cu(111), with the aim to understand their effects on the $*NO_3$ and $*N$ binding energies. A significant finding is that by tailoring the interatomic coupling strength (βV_{ad}^2) from a subsurface metal ligand to an extent it is possible to break the linear scaling between $*N$ and $*NO_3$

adsorption energies on {100} site ensembles. Consistent with the *d*-band theory, the site electronic structure (e.g., the *d*-center) influences *NO₃ and *N adsorption energies in a similar way and its variation can not be translated into breaking scaling. Those machine-learned insights are leveraged to dramatically reduce the design space of intermetallic catalysts, allowing us to specifically focus on {100}-terminated body centered cubic nanostructures which exhibit beyond-scaling characteristics based on structural analysis in Supplementary Fig. 8. In those systems, the subsurface metal ligand is in the first-neighbor shell of the hollow *N adsorbate, thus greatly increasing the interatomic coupling strength between the metal *d*-states and N 2p states. With this theoretical guidance, we performed DFT calculations of *NO₃ and *N binding energies on this subset of the materials space. The results of this screening are shown in Figure 2 in which several systems (e.g., CuPd, ZnCu, and ZnRh) exhibit beyond-scaling reactivity properties, justifying the ML-guided methodology. In terms of the accuracy, DFT-predicted ΔG^{\max} of CuPd(100) at Cu sites is 0.101 eV lower than that at Cu(100). Although DFT calculations at the GGA-PBE level have an often-quoted error of ± 0.2 eV for adsorption energies at metal surfaces, the relative error across similar systems is expected to be much smaller and the qualitative prediction is reliable in the methodology.

Changes: We added the following into the main text: “DFT-calculated free energy diagrams of the full reaction pathway predict that the activity metric ΔG^{\max} at CuPd(100) is 0.101 eV lower than that at Cu(100), with both surfaces limited by *NO₃ adsorption (Supplementary Fig. 9). Although DFT calculations at the GGA-PBE level have an often-quoted error of ± 0.2 eV for adsorption energies at metal surfaces, the relative error across similar systems is expected to be much smaller and the qualitative prediction is reliable in the methodology.”

Q2. Given the results predicting that CuPd will be near to the top of the activity volcano, I was expecting the authors to report a deeper analysis of the electronic structure of CuPd as it relates to N* and NO₃* binding. However, the authors only confirm their prediction with a DFT reaction energy diagram in S. Figure 8. The manuscript would be greatly enhanced if the authors were to include a much deeper analysis of the CuPd system at the electronic level (presented side by side with pure Cu for comparison), which will clearly illustrate how NO₃* and N* adsorption are independently tuned by the underlying electronic structure.

Response: We thank the reviewer for this comment. The detailed electronic structure analysis is important to validate the physical understanding of chemical bonding attained from machine learning. Supplementary Figure 10a and 10b show the projected density of states onto the N 2p and surface Cu 3d orbitals for *N on Cu(100) and CuPd(100), respectively. Dashed lines represent the *d*-band center of a surface Cu atom. We can see that the Cu *d*-band center of CuPd(100) is higher in energy than that of Cu(100). According to the *d*-band theory, adsorbates would likely bind stronger on CuPd than on Cu because of a decreased population of adsorbate-metal antibonding states that is a consequence of a higher *d*-band center. This explains why the bidentate *NO₃ binds stronger on CuPd than on Cu. However, this is not the case for the hollow *N species. On CuPd(100), the increase in the attractive orbital hybridization energy (more exothermic) due

to a higher d -band center can be surpassed by an increase in the repulsive orbital orthogonalization energy (more endothermic) because of a larger d -orbital radius of Pd than Cu and a shorter bond distance of the subsurface Pd to the hollow *N adsorbate. Since the interatomic coupling strength (βV_{ad}^2) largely controls the energy gap between bonding and antibonding states. A stronger (weaker) interatomic coupling leads to a larger (smaller) energy separation of bonding and antibonding states. This can be evidenced in Supplementary Figure 10a and b showing projected density of states onto *N for Cu(100) and CuPd(100), respectively. This increase in interatomic coupling affects the hybridization and repulsive contributions, albeit to a different extent. From the energy decomposition analysis in Supplementary Figure 10c using the *Bayeschem* model, we found that the repulsive contribution dominates the overall interaction between the *N 2p orbitals and metal d -states for most of the metals including Cu and Pd. Ultimately, this bonding mechanism makes it possible to realize an independent tuning of the *N and *NO₃ binding energies, where the higher d -center of site atoms stabilizes *NO₃ while the larger interatomic coupling from the subsurface metal ligand destabilizes the *N.

Supplementary Figure 10 | Electronic structure analysis of *N on Cu(100) and CuPd(100).

a, Density of states projected onto the p_{xy} and p_z orbitals of *N on Cu(100) and the d -states of a surface Cu atom on clean Cu(100). b, Density of states projected onto the p_{xy} and p_z orbitals of *N on CuPd(100) and the d -states of a surface Cu atom on clean CuPd(100). The dashed line represents the d -center, and peaks of adsorbate-metal bonding and antibonding states are displayed via markers. c, Decomposition of the adsorption energy associated with metal d -states into orbital hybridization and repulsion contributions using the *Bayeschem* ML model for *N on M(100) surfaces. The sp -band contribution is assumed to be constant in the *Bayeschem* ML model.

Changes: We have included Supplementary Fig. 10 in the SI and added the following discussion in the main text: “To validate the physical understanding of chemical bonding attained from machine learning, we have performed detailed electronic structure analysis of Cu(100) and CuPd(100) in Supplementary Fig. 10. As suggested by the Bayeschem ML models, the higher *d*-center of site Cu atoms at CuPd stabilizes *NO₃ while the larger interatomic coupling from the subsurface Pd ligand destabilizes the *N, realizing an independent tuning of the *N and *NO₃ binding energies to a certain extent.”.

Q3. On Pg. 3 the authors state “Single-crystal experiments showed that NO₃RR on Cu is structure sensitive with (100)-oriented surface sites more active toward NH₃ formation than the (111) counterparts.” I believe this sentence should refer to reference 5 (Koper and co-workers) and not reference 6.

Response: Thanks for the reviewer’s kind suggestion. We have changed the corresponding reference in the revised manuscript.

Changes: We have changed the corresponding reference from reference 6 to reference 5.

Q4. On Pg. 10 the authors state “DFT calculations of reaction pathways confirmed the model prediction with CuPd limited by *NO₃ adsorption and Cu termination is more stable than Pd termination (Supplementary Fig. 8).” I don’t see anything in this figure regarding the stability of the Cu termination versus the Pd termination. How was this stability difference determined?

Response: We thank the reviewer for pointing this out. In Supplementary Fig. 9 in the revised manuscript, we show that CuPd(100) is more active than Cu(100) for nitrate reduction to NH₃ based on DFT-calculated energetics of the reaction pathway. For thermodynamic stability of Cu and Pd terminations of CuPd(100), we were referring to previous studies (Jiang, Y.; Li, H.; Wu, Z.; Ye, W.; Zhang, H.; Wang, Y.; Sun, C.; Zhang, Z. In Situ Observation of Hydrogen-Induced Surface Faceting for Palladium-Copper Nanocrystals at Atmospheric Pressure. *Angew. Chem. Int. Ed Engl.* **2016**, *55* (40), 12427–12430; Tang, M.; Li, H.; Yuan, W.; Zou, S.; Sun, C.; Wang, Y. First-Principles Study of the Interactions of Hydrogen with Low-Index Surfaces of PdCu Ordered Alloy. *Progress in Natural Science: Materials International* **2017**, *27* (6), 709–713.) showing that the Cu termination is more stable than Pd termination on both (100) and (111) facets of B2 CuPd intermetallic nanostructures.

Changes: We added the following discussion with references for the surface stability: “Cu-terminated CuPd (100) and (111) surfaces have been previously shown to be more stable than the Pd-termination using DFT-calculated surface energies^{31,32}.”

Q5. On Pg. 6 the authors state “Moreover, *NO₂ formation from NO₂⁻ is more exergonic on Cu(111) than that on Cu(100), resulting in a slower further reduction of NO₂⁻ on (111)-type sites as observed.” I do not follow why this would lead to slower reduction of NO₂⁻ on (111) sites. Please clarify the logic here. Did the authors mean to say that NO₂⁻ formation from NO₂* is more exergonic on Cu(111), and thus NO₂⁻ desorbs more readily from (111) and therefore reduction

beyond NO₂⁻ occurs less readily on (111)? If so, it is interesting to note that a similar effect was reported for Pd(111) versus Pd(100) in Hatzell et al. ACS Catal. 2021, 11, 12, 7568–7577.

Response: What we mean further reduction of NO₂⁻ should be re-adsorption and further reduction of NO₂⁻. That would be clear. Thanks for pointing this out.

Changes: We have cited this paper in the revised manuscript. And we changed the sentence “Moreover, *NO₂ formation from NO₂⁻ is more exergonic on Cu(111) than that on Cu(100), resulting in a slower further reduction of NO₂⁻ on (111)-type sites as observed.” To “Moreover, *NO₂ formation from NO₂⁻ is more exergonic on Cu(111) than that on Cu(100), resulting in a slower re-adsorption of NO₂⁻ and its further reduction on (111)-type sites as observed.”

Q6. In the SI on Ln. 72 the authors state “An additional intermediate dependent constant was added to account for the electrolyte/electrode interface and is tabulated in the last column of Supplementary Table 1.” Can the authors clarify how this correction was determined? It appears that the authors use GC-DFT on Cu(100) to derive this correction value, and then assume that the same value can be used for all surfaces. If this is the case, can the authors include benchmarking tests on an additional surface (or two) to quantify error associated with such an assumption?

Response: Yes. The electrode potential correction to the free formation energy of a surface intermediate was obtained from GC-DFT calculations on Cu(100) at 0 V vs RHE. It was assumed that this correction is approximately constant across metal surfaces at this potential. To re-assess this assumption, we performed GC-DFT calculations of all relevant intermediates on Pt(100), Pd(100), Au(100) and Ag(100). Supplementary Table 2 tabulates the correction value for each surface and the average of all five surfaces. We can see that two largest average corrections are +0.568 and +0.363 eV for *NO₃ and *NO₂, respectively. Those values are very close (< 0.05 eV) to the Cu(100) values that we used (0.529 and 0.320 eV).

Changes: We included the following Supplementary Table 2 in SI and added the following into the DFT method section in the main text. “The electrode potential correction to the free formation energy of a surface intermediate was obtained from GC-DFT calculations on Cu(100) at 0 V vs RHE. This correction is assumed to be constant, which is a reasonable approximation across metal surfaces at this potential (Supplementary Table 2).”

Supplementary Table 2 | Energy corrections due to the constant electrode potential. For all adsorbates and selected surfaces, GC-DFT calculations were performed to quantify the effect of solvation and electrochemical bias at 0 V vs. RHE and a pH of 14 on the free formation energies of reaction intermediates. The reported values are the difference in formation free energies calculated using GC-DFT energetics and standard DFT energetics. The last column is the average energy correction from all the surfaces.

Adsorbate	Pt(100)	Pd(100)	Au(100)	Ag(100)	Cu(100)	Average (eV)
*NO ₃	0.416	0.632	0.539	0.726	0.529	0.568
*NO ₂	0.262	0.370	0.487	0.378	0.320	0.363

*NO	0.005	0.012	0.054	0.078	0.192	0.068
*NHO	-0.196	-0.052	0.289	0.129	0.128	0.060
*NH ₂ O	-0.174	0.030	0.127	0.164	0.148	0.059
*NH ₂ OH	-.224	-.007	0.020	0.193	0.137	0.024
*NH ₂	-.306	-.120	-.102	0.022	0.02	-.097

Q7. In the SI on Ln. 52 the authors state “To get the Gibbs free energy of the alkaline reaction at a given pH, an additional correction $0.059 \times \text{pH}$ was added (+1.22 eV at pH 14 in our system).” I suggest that the authors re-word this and state that the correction at pH 14 is 0.83 eV, which summed with the value in the preceding sentence of 0.39 eV equals a total correction of +1.22 eV. **Response:** We have changed “To get the Gibbs free energy of the alkaline reaction at a given pH, an additional correction $0.059 \times \text{pH}$ was added (+1.22 eV at pH 14 in our system).” To “To get the Gibbs free energy of the above reaction and its alkaline counterpart at a given pH, an additional correction $0.059 \times \text{pH}$ was added (e.g., $0.392+0.0592 \times 14 = +1.22$ eV at pH 14 in our system).”

Reviewer #2 (Remarks to the Author):

This work proposes ordered B2 CuPd nanocubes that demonstrate exceedingly high performance for NO₃RR to ammonia with a FE of 92.5% at -0.5 VRHE and a yield rate of 6.25 mol h⁻¹ g⁻¹ at -0.6 VRHE. This study provides machine-learned design rules that can be realized on (100)-type sites of ordered B2 intermetallics, in which the orbital overlap between the hollow *N and subsurface metal atoms is significant while the bridge-bidentate *NO₃ is not affected. However, some important discussion and information are missing such as the investigation about the origin of activities are insufficient. I recommend reconsidering this work after a major revision. Please see the comments below.

Response: We thank the reviewer for his/her recommendation and constructive comments. We have addressing the raised comments point by point as follows:

Q1. The author insisted that the conventional method for synthesis of intermetallic structure has a disadvantage for sintering or preserving the nanocrystals, but the solution-phase synthesis author used, the NP size is already large over 20 nm and the low-resolution TEM (Supplementary Figure 26) showed the nanocrystal shapes are collapsed. It's hard to be sure what the benefits of this method are.

Response: Thanks for the reviewer’s comment. We agree with the reviewer that the size of the CuPd nanocubes is larger than 20 nm. Fortunately, the ordered CuPd cubes are uniform and nearly monodispersed. We also would like to emphasize that the approach we developed here is a direct, one-pot synthesis of monodisperse ordered intermetallic nanocrystals in the solution phase. Previously, it always requires an additional step either electrochemically, through thermal annealing, or a seed-mediated growth and diffusion process, and the produced intermetallic alloys

are partially ordered. In addition, we have provided the high-resolution TEM image of the carbon-supported ordered CuPd nanocubes after the stability test in the revised manuscript. After the stability test, the cubic morphology was mostly maintained while the corners of the nanocubes were slightly roughened due to the prolonged electrochemical reaction, and there was no significant aggregation nor size change. The HRTEM image (**Supplementary Fig. 32c**) shows clear lattice fringes with a lattice spacing of 2.95 Å, which can be assigned to the (100) planes of the ordered CuPd intermetallic phase, suggesting the structure of CuPd maintains after long-time electrolysis. In summary, our approach produces well-defined, uniform intermetallic CuPd nanocubes directly in one-pot solution synthesis without additional treatment, which provides a basis to validate our computational predictions. Meanwhile, those ordered CuPd nanocubes maintain their size, structure, and most of their morphology after long-time electrolysis.

Supplementary Figure 32 | Stability test. a, b, TEM images and c, HRTEM image of carbon-supported ordered CuPd nanocubes after stability test by running the CA measurement at -0.5 V vs. RHE for 12 h.

Changes: We added Supplementary Figure 32 into SI and added the following into the main text: “The HRTEM image (Supplementary Fig. 32c) shows clear lattice fringes with a lattice spacing of 2.95 Å, which can be assigned to the (100) planes of the ordered CuPd intermetallic phase.”

Q2. The applied potential -0.5 VRHE is quite negative and low. This potential range is enough to produce much hydrogen, but authors provide 92.5% FE for ammonia. Pd is also well-known material for high N₂ selectivity, but there is no information about N₂ FE. Also, more negative potentials should produce more hydrogen but why the hydrogen FE is getting lower in Supplementary figure 23b, d, f?

Response: Thanks for the reviewer’s comments and questions. Although Pd is a well-known material for high N₂ selectivity in neutral media, it is more likely to produce ammonia in a highly alkaline medium (*Acs Catal.* 2021, 11, 7568-7577, *Nanoscale* 2021, 13, 17504-17511, *Energy Environ. Sci.* 2021, 14, 3938-3944). The direct N-N coupling to yield N₂ is kinetically less favorable than the combination of a highly mobile H* with N* to yield NH₃. In our case, we did not observe significant N₂ production from the NO₃⁻ reduction.

For the question regarding HER, it is well known that a highly alkaline medium can suppress HER (*J. Am. Chem. Soc.* 2020, 142, 7036-7046, *Nat. Commun.* 2019, 10, 631). Meanwhile, in the ordered intermetallic CuPd nanocubes, Cu is the active site for the adsorption of NO₃* while Pd might be the binding site of H*. By introducing Pd, the ordered CuPd structure favors H* formation more than pure Cu surfaces. However, instead of forming H₂, we found under our testing conditions (highly alkaline) on CuPd surfaces, the resulting H* promotes the hydrogenation of reaction intermediates from NO₃RR to yield ammonia. At more negative potentials, although more H* produced, there are also much more reaction intermediates from NO₃RR that need to be hydrogenated. Therefore, at more negative potentials, the FE of ammonia increases while the hydrogen FE decrease, which can be attributed to a favored coupling between *H and NO₃RR intermediates over the *H-H coupling.

Q3. Please provide B2 intermetallic CuPd/C catalysts pristine of TEM information. Supplementary Figure 26 showed only after stability test TEM data, no pristine data for carbon supported CuPd catalysts. Then readers can compare the stability test data. Also, it is hard to verify the intermetallic structure of CuPd nanocrystals is remained because of very low-resolution. But it seems the shapes are collapsed. Please provide the high-resolution TEM image after the test to prove the intermetallic structure remains. Not only this, change of chemical compositions, states (XPS for Supplementary Figure 27b has a poor quality for discussing), leaching concentrations are missing to discuss the stability. After the test, the lower FE of 85.1% is measured. Please provide the discussion for this.

Response: Thanks for the reviewer's suggestions and comments. In the revised manuscript, we have provided the TEM image of pristine carbon-supported ordered CuPd nanocubes as well as the high-resolution TEM image of the carbon-supported ordered CuPd nanocubes after the stability test. After the stability test, the size was maintained while the corners of the nanocubes were slightly roughened due to the prolonged electrochemical reaction. The HRTEM image (**Supplementary Fig. 32c**) shows clear lattice fringes with a lattice spacing of 2.95 Å, which can be assigned to the (100) planes of the ordered CuPd intermetallic phase, suggesting the structure of CuPd maintained after long-time electrolysis. We also performed XRD measurement after stability test (**Supplementary Fig. 33**), which showed that the CuPd nanocubes still maintain the ordered *B2* phase after long-term electrolysis. Besides, the ICP-MS result shows that the Pd/Cu molar ratio is still close to 1:1. Although the XPS spectra (**Supplementary Fig. 34**) are noisy because of the small amount catalyst loaded on carbon used for the stability test, it is still clear to see that there is no obvious peak shift after the stability test. There is a slight decrease in the NO₃RR electrocatalytic performance after the stability test, possibly due to a small fraction of the catalyst falling off from the carbon paper.

Supplementary Figure 22 | TEM image. a, TEM image of carbon-supported ordered CuPd nanocubes.

Supplementary Figure 32 | Stability test. a, b, TEM images and c, HRTEM image of carbon-supported ordered CuPd nanocubes after stability test by running the CA measurement at -0.5 V vs. RHE for 12 h.

Supplementary Figure 33 | Stability test. XRD pattern of carbon-supported ordered CuPd nanocubes after stability test.

Changes: We added Supplementary Fig. 22, Supplementary Fig. 32, and Supplementary Fig. 33 into SI and added the following into the main text: “There is a slight decrease of electrocatalytic performance after the stability test, possibly due to a small fraction of the catalyst falling off from the carbon paper.” and “The HRTEM image (Supplementary Fig. 32c) shows clear lattice fringes with a lattice spacing of 2.95 Å, which can be assigned to the (100) planes of the ordered CuPd intermetallic phase. Moreover, the ICP-MS result shows that the Pd/Cu molar ratio is still close to 1:1. We also performed XRD measurement after stability test (Supplementary Fig. 33), which showed that the CuPd nanocubes still maintain the ordered *B2* phase after long-term electrolysis.”

Q4. Please provide HER LSV curves of all catalysts in Figure 5a for reference.

Response: Thanks for the reviewer’s suggestion. According to your suggestion, we have provided HER LSV curves of all catalysts in Figure 5a for reference.

Supplementary Fig. 25 | Electrocatalytic HER performance. LSV curves of ordered CuPd nanocubes, Cu nanocubes, and Pd nanocubes normalized to the geometric area. The polarization curves were obtained by sweeping the potential from -0.5 to -1.7 V vs. Ag/AgCl with a sweep rate of 20 mV s^{-1} in 1 M KOH.

Changes: We added **Supplementary Fig. 25** into SI and added the following into the main text: “As shown in **Supplementary Fig. 25**, the ordered CuPd nanocubes show higher HER activity than Cu nanocubes while Pd nanocubes perform best among three tested.”

Q5. Please provide the quantification information for NO_3^- and NO_2^- with a curve of UV-vis spectra and wavelength.

Response: Thanks for the reviewer’s suggestion. We have provided the ultraviolet-visible adsorption spectra of different solution with different ammonium concentrations and the corresponding calibration curve (**Supplementary Fig. 29**). We did not use UV-vis spectra to quantify the concentrations of NO_2^- . Instead, we used ion chromatography to detect NO_2^- , and we have provided the calibration curve of NO_2^- with different concentrations (**Supplementary Fig. 30**).

Supplementary Figure 29 | UV-vis measurements of the concentrations of ammonium with Nessler’s reagents. (a) The ultraviolet-visible absorption spectra of different solutions with different ammonium concentrations. (b) Calibration curve for colorimetric NH₃ assay using Nessler reagent. Electrolytes after catalysis were diluted to ensure the ammonia concentrations in the test solutions were in the linear range of the calibration curve.

Supplementary Figure 30 | Ion chromatography measurements of the concentrations of NO₂⁻. Calibration curve of NO₂⁻ with different concentrations. Electrolytes after catalysis were diluted to ensure the NO₂⁻ concentrations in the test solutions were in the linear range of the calibration curve.

Changes: We added Supplementary Figure 29 and Supplementary Figure 30 into SI.

REVIEWERS' COMMENTS

Reviewer #1 (Remarks to the Author):

The authors did a good job responding to my comments and revising the manuscript accordingly. This work is now suitable for publication.

Reviewer #2 (Remarks to the Author):

The revised manuscript sufficiently addressed my concerns.

Reviewer #3 (Remarks to the Author):

The authors have addressed the suggestions and the paper is now suitable for publication.